# Quantifying and addressing the prevalence and bias of study designs in the environmental and social sciences

Alec P. Christie ⓘ et al.#

Building trust in science and evidence-based decision-making depends heavily on the credibility of studies and their findings. Researchers employ many different study designs that vary in their risk of bias to evaluate the true effect of interventions or impacts. Here, we empirically quantify, on a large scale, the prevalence of different study designs and the magnitude of bias in their estimates. Randomised designs and controlled observational designs with pre-intervention sampling were used by just 23% of intervention studies in biodiversity conservation, and 36% of intervention studies in social science. We demonstrate, through pairwise within-study comparisons across 49 environmental datasets, that these types of designs usually give less biased estimates than simpler observational designs. We propose a model-based approach to combine study estimates that may suffer from different levels of study design bias, discuss the implications for evidence synthesis, and how to facilitate the use of more credible study designs.

---

#A list of authors and their affiliations appears at the end of the paper.

The ability of science to reliably guide evidence-based decision-making hinges on the accuracy and credibility of studies and their results[1,2]. Well-designed, randomised experiments are widely accepted to yield more credible results than non-randomised, 'observational studies' that attempt to approximate and mimic randomised experiments[3]. Randomisation is a key element of study design that is widely used across many disciplines because of its ability to remove confounding biases (through random assignment of the treatment or impact of interest[4,5]. However, ethical, logistical, and economic constraints often prevent the implementation of randomised experiments, whereas non-randomised observational studies have become popular as they take advantage of historical data for new research questions, larger sample sizes, less costly implementation, and more relevant and representative study systems or populations[6–9]. Observational studies nevertheless face the challenge of accounting for confounding biases without randomisation, which has led to innovations in study design.

We define 'study design' as an organised way of collecting data. Importantly, we distinguish between data collection and statistical analysis (as opposed to other authors[10]) because of the belief that bias introduced by a flawed design is often much more important than bias introduced by statistical analyses. This was emphasised by Light, Singer & Willet[11] (p. 5): "You can't fix by analysis what you bungled by design…"; and Rubin[3]: "Design trumps analysis." Nevertheless, the importance of study design has often been overlooked in debates over the inability of researchers to reproduce the original results of published studies (so-called 'reproducibility crises'[12,13]) in favour of other issues (e.g., p-hacking[14] and Hypothesizing After Results are Known or 'HARKing'[15]).

To demonstrate the importance of study designs, we can use the following decomposition of estimation error equation[16]:

$$\text{Estimation error} = (\text{Estimator} - \text{true causal effect})$$
$$= (\text{Design bias} + \text{Modelling bias} + \text{Statistical noise}). \quad (1)$$

This demonstrates that even if we improve the quality of modelling and analysis (to reduce modelling bias through a better bias-variance trade-off[17]) or increase sample size (to reduce statistical noise), we cannot remove the intrinsic bias introduced by the choice of study design (design bias) unless we collect the data in a different way. The importance of study design in determining the levels of bias in study results therefore cannot be overstated.

For the purposes of this study we consider six commonly used study designs; differences and connections can be visualised in Fig. 1. There are three major components that allow us to define these designs: randomisation, sampling before and after the impact of interest occurs, and the use of a control group.

Of the non-randomised observational designs, the Before-After Control-Impact (BACI) design uses a control group and samples before and after the impact occurs (i.e., in the 'before-period' and the 'after-period'). Its rationale is to explicitly account for pre-existing differences between the impact group (exposed to the impact) and control group in the before-period, which might otherwise bias the estimate of the impact's true effect[6,18,19].

The BACI design improves upon several other commonly used observational study designs, of which there are two uncontrolled designs: After, and Before-After (BA). An After design monitors an impact group in the after-period, while a BA design compares the state of the impact group between the before- and after-periods. Both designs can be expected to yield poor estimates of the impact's true effect (large design bias; Equation (1)) because

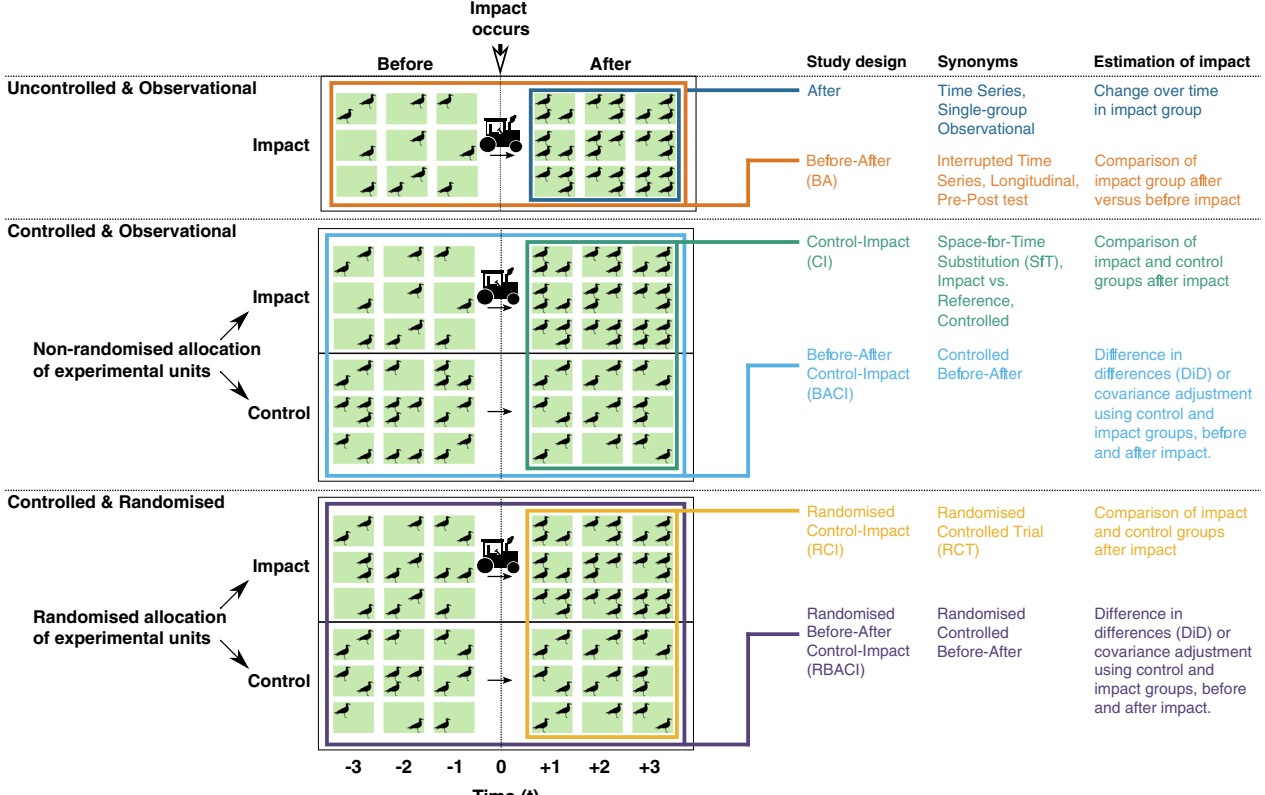

**Fig. 1 Comparison of different study designs used to evaluate the effect of an impact.** A hypothetical study set-up is shown where the abundance of birds in three impact and control replicates (e.g., fields represented by blocks in a row) are monitored before and after an impact (e.g., ploughing) that occurs in year zero. Different colours represent each study design and illustrate how replicates are sampled. Approaches for calculating an estimate of the true effect of the impact for each design are also shown, along with synonyms from different disciplines.

changes in the response variable could have occurred without the impact (e.g., due to natural seasonal changes; Fig. 1).

The other observational design is Control-Impact (CI), which compares the impact group and control group in the after-period (Fig. 1). This design may suffer from design bias introduced by pre-existing differences between the impact group and control group in the before-period; bias that the BACI design was developed to account for[20,21]. These differences have many possible sources, including experimenter bias, logistical and environmental constraints, and various confounding factors (variables that change the propensity of receiving the impact), but can be adjusted for through certain data pre-processing techniques such as matching and stratification[22].

Among the randomised designs, the most commonly used are counterparts to the observational CI and BACI designs: Randomised Control-Impact (R-CI) and Randomised Before-After Control-Impact (R-BACI) designs. The R-CI design, often termed 'Randomised Controlled Trials' (RCTs) in medicine and hailed as the 'gold standard'[23,24], removes any pre-impact differences in a stochastic sense, resulting in zero design bias (Equation (1)). Similarly, the R-BACI design should also have zero design bias, and the impact group measurements in the before-period could be used to improve the efficiency of the statistical estimator. No randomised equivalents exist of After or BA designs as they are uncontrolled.

It is important to briefly note that there is debate over two major statistical methods that can be used to analyse data collected using BACI and R-BACI designs, and which is superior at reducing modelling bias[25] (Equation (1)). These statistical methods are: (i) Differences in Differences (DiD) estimator; and (ii) covariance adjustment using the before-period response, which is an extension of Analysis of Covariance (ANCOVA) for generalised linear models — herein termed 'covariance adjustment' (Fig. 1). These estimators rely on different assumptions to obtain unbiased estimates of the impact's true effect. The DiD estimator assumes that the control group response accurately represents the impact group response had it not been exposed to the impact ('parallel trends'[18,26]) whereas covariance adjustment assumes there are no unmeasured confounders and linear model assumptions hold[6,27].

From both theory and Equation (1), with similar sample sizes, randomised designs (R-BACI and R-CI) are expected to be less biased than controlled, observational designs with sampling in the before-period (BACI), which in turn should be superior to observational designs without sampling in the before-period (CI) or without a control group (BA and After designs)[7,28]. Between randomised designs, we might expect that an R-BACI design performs better than a R-CI design because utilising extra data before the impact may improve the efficiency of the statistical estimator by explicitly characterising pre-existing differences between the impact group and control group.

Given the likely differences in bias associated with different study designs, concerns have been raised over the use of poorly designed studies in several scientific disciplines[7,29–35]. Some disciplines, such as the social and medical sciences, commonly undertake direct comparisons of results obtained by randomised and non-randomised designs within a single study[36–38] or between multiple studies (between-study comparisons[39–41]) to specifically understand the influence of study designs on research findings. However, within-study comparisons are limited in their scope (e.g., a single study[42,43]) and between-study comparisons can be confounded by variability in context or study populations[44]. Overall, we lack quantitative estimates of the prevalence of different study designs and the levels of bias associated with their results.

In this work, we aim to first quantify the prevalence of different study designs in the social and environmental sciences. To fill this knowledge gap, we take advantage of summaries for several thousand biodiversity conservation intervention studies in the Conservation Evidence database[45] (www.conservationevidence.com) and social intervention studies in systematic reviews by the Campbell Collaboration (www.campbellcollaboration.org). We then quantify the levels of bias in estimates obtained by different study designs (R-BACI, R-CI, BACI, BA, and CI) by applying a hierarchical model to approximately 1000 within-study comparisons across 49 raw environmental datasets from a range of fields. We show that R-BACI, R-CI and BACI designs are poorly represented in studies testing biodiversity conservation and social interventions, and that these types of designs tend to give less biased estimates than simpler observational designs. We propose a model-based approach to combine study estimates that may suffer from different levels of study design bias, discuss the implications for evidence synthesis, and how to facilitate the use of more credible study designs.

## Results

**Prevalence of study designs.** We found that the biodiversity-conservation (conservation evidence) and social-science (Campbell collaboration) literature had similarly high proportions of intervention studies that used CI designs and After designs, but low proportions that used R-BACI, BACI, or BA designs (Fig. 2). There were slightly higher proportions of R-CI designs used by intervention studies in social-science systematic reviews than in the biodiversity-conservation literature (Fig. 2). The R-BACI, R-CI, and BACI designs made up 23% of intervention studies for biodiversity conservation, and 36% of intervention studies for social science.

**Influence of different study designs on study results.** In non-randomised datasets, we found that estimates of BACI (with covariance adjustment) and CI designs were very similar, while the point estimates for most other designs often differed substantially in their magnitude and sign. We found similar results in randomised datasets for R-BACI (with covariance adjustment) and R-CI designs. For ~30% of responses, in both non-randomised and randomised datasets, study design estimates differed in their statistical significance (i.e., p < 0.05 versus p > =0.05), except for estimates of (R-)BACI (with covariance adjustment) and (R-)CI designs (Table 1; Fig. 3). It was rare for the 95% confidence intervals of different designs' estimates to not overlap – except when comparing estimates of BA designs to (R-)BACI (with covariance adjustment) and (R-)CI designs (Table 1). It was even rarer for estimates of different designs to have significantly different signs (i.e., one estimate with entirely negative confidence intervals versus one with entirely positive confidence intervals; Table 1, Fig. 3). Overall, point estimates often differed greatly in their magnitude and, to a lesser extent, in their sign between study designs, but did not differ as greatly when accounting for the uncertainty around point estimates – except in terms of their statistical significance.

**Levels of bias in estimates of different study designs.** We modelled study design bias using a random effect across datasets in a hierarchical Bayesian model; σ is the standard deviation of the bias term, and assuming bias is randomly distributed across datasets and is on average zero, larger values of σ will indicate a greater magnitude of bias (see Methods). We found that, for randomised datasets, estimates of both R-BACI (using covariance adjustment; CA) and R-CI designs were affected by negligible amounts of bias (very small values of σ; Table 2). When the R-BACI design used the DiD estimator, it suffered from slightly more bias (slightly larger values of σ), whereas the BA design had

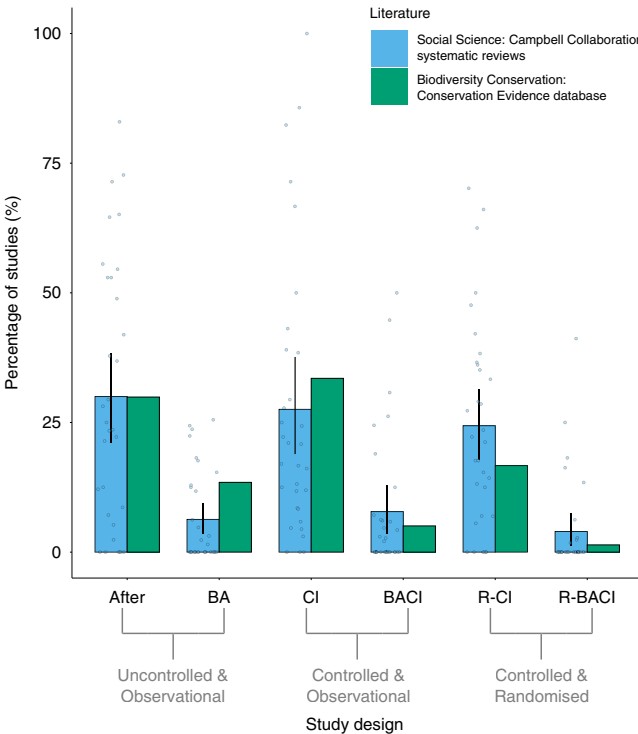

**Fig. 2 Percentage of intervention studies with different study designs in the biodiversity-conservation and social-science literature.** Intervention studies from the biodiversity-conservation literature were screened from the Conservation Evidence database (*n*=4260 studies) and studies from the social-science literature were screened from 32 Campbell Collaboration systematic reviews (*n*=1009 studies – note studies excluded by these reviews based on their study design were still counted). Percentages for the social-science literature were calculated for each systematic review (blue data points) and then averaged across all 32 systematic reviews (blue bars and black vertical lines represent mean and 95% Confidence Intervals, respectively). Percentages for the biodiversity-conservation literature are absolute values (shown as green bars) calculated from the entire Conservation Evidence database (after excluding any reviews). Source data are provided as a Source Data file. BA before-after, CI control-impact, BACI before-after-control-impact, R-BACI randomised BACI, R-CI randomised CI.

very high bias when applied to randomised datasets (very large values of σ; Table 2). There was a highly positive correlation between the estimates of R-BACI (using covariance adjustment) and R-CI designs (Ω[R-BACI CA, R-CI] was close to 1; Table 2). Estimates of R-BACI using the DiD estimator were also positively correlated with estimates of R-BACI using covariance adjustment and R-CI designs (moderate positive mean values of Ω[R-BACI CA, R-BACI DiD] and Ω[R-BACI DiD, R-CI]; Table 2).

For non-randomised datasets, controlled designs (BACI and CI) were substantially less biased (far smaller values of σ) than the uncontrolled BA design (Table 2). A BACI design using the DiD estimator was slightly less biased than the BACI design using covariance adjustment, which was, in turn, slightly less biased than the CI design (Table 2).

Standard errors estimated by the hierarchical Bayesian model were reasonably accurate for the randomised datasets (see λ in Methods and Table 2), whereas there was some underestimation of standard errors and lack-of-fit for non-randomised datasets.

## Discussion

Our approach provides a principled way to quantify the levels of bias associated with different study designs. We found that randomised study designs (R-BACI and R-CI) and observational BACI designs are poorly represented in the environmental and social sciences; collectively, descriptive case studies (the After design), the uncontrolled, observational BA design, and the controlled, observational CI design made up a substantially greater proportion of intervention studies (Fig. 2). And yet R-BACI, R-CI and BACI designs were found to be quantifiably less biased than other observational designs.

As expected the R-CI and R-BACI designs (using a covariance adjustment estimator) performed well; the R-BACI design using a DiD estimator performed slightly less well, probably because the differencing of pre-impact data by this estimator may introduce additional statistical noise compared to covariance adjustment, which controls for these data using a lagged regression variable. Of the observational designs, the BA design performed very poorly (both when analysing randomised and non-randomised data) as expected, being uncontrolled and therefore prone to severe design bias[7,28]. The CI design also tended to be more biased than the BACI design (using a DiD estimator) due to pre-existing differences between the impact and control groups. For BACI designs, we recommend that the underlying assumptions of

---

**Table 1 Pairwise comparison of estimates obtained using different study designs.**

| Design 1 | Design 2 | No overlap (95% Conf. Ints.) | >100% difference in magnitude (P.E.) | Different significance (95% Conf. Ints.) | Different signs (P.E.) | Significantly different sign (95% Conf. Ints.) |
|---|---|---|---|---|---|---|
| Randomised (R-) | | | | | | |
| BACI DiD | BACI CA | 0.01 | 0.68 | 0.27 | 0.32 | 0.00 |
| BACI DiD | CI | 0.01 | 0.69 | 0.27 | 0.32 | 0.00 |
| BACI DiD | BA | 0.01 | 0.68 | 0.29 | 0.34 | 0.00 |
| BACI CA | CI | 0.00 | 0.04 | 0.05 | 0.01 | 0.00 |
| BACI CA | BA | 0.16 | 0.82 | 0.33 | 0.47 | 0.06 |
| CI | BA | 0.16 | 0.82 | 0.30 | 0.47 | 0.07 |
| Non-randomised | | | | | | |
| BACI DiD | BACI CA | 0.04 | 0.58 | 0.31 | 0.27 | 0.00 |
| BACI DiD | CI | 0.05 | 0.61 | 0.28 | 0.30 | 0.01 |
| BACI DiD | BA | 0.04 | 0.61 | 0.22 | 0.25 | 0.01 |
| BACI CA | CI | 0.00 | 0.18 | 0.08 | 0.08 | 0.00 |
| BACI CA | BA | 0.14 | 0.74 | 0.34 | 0.36 | 0.03 |
| CI | BA | 0.12 | 0.71 | 0.33 | 0.37 | 0.02 |

This shows the proportion of responses in which there were differences in the magnitude (by > 100%) and sign of estimates, and differences in the significance, sign and overlap between associated 95% confidence intervals. For randomised datasets, BACI and CI labels refer to R-BACI and R-CI designs (denoted by 'R-'). The 100% difference in magnitude criterion is set relative to the smaller estimate. BA before-after, BACI before-after-control-impact, CI control-impact, DiD difference in differences, CA covariance adjustment, 95% Conf. Ints. refers to 95% confidence intervals, P.E. point estimate.

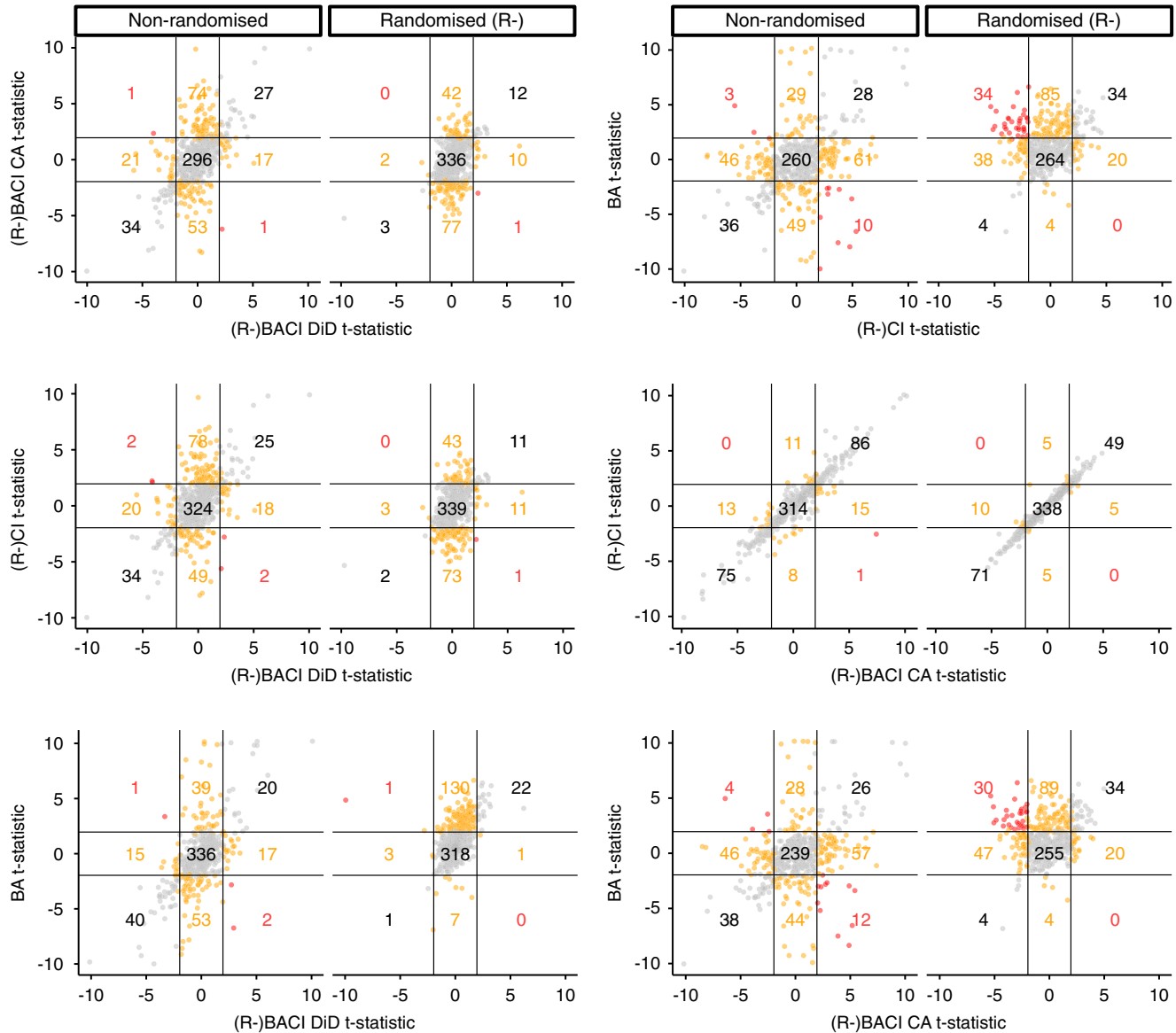

**Fig. 3 Pairwise comparisons of t-statistics for estimates obtained using different study designs for responses across 49 different datasets (non-randomised or randomised).** t-statistics were obtained from two-sided t-tests of estimates obtained by each design for different responses in each dataset using Generalised Linear Models (see Methods). For randomised datasets, BACI and CI axis labels refer to R-BACI and R-CI designs (denoted by 'R-'). DiD Difference in Differences; CA covariance adjustment. Lines at t-statistic values of 1.96 denote boundaries between cells and colours of points indicate differences in direction and statistical significance ($p < 0.05$; grey = same sign and significance, orange = same sign but difference in significance, red = different sign and significance). Numbers refer to the number of responses in each cell. Source data are provided as a Source Data file. BA Before-After, CI Control-Impact, BACI Before-After-Control-Impact.

DiD and CA estimators are carefully considered before choosing to apply them to data collected for a specific research question[6,27]. Their levels of bias were negligibly different and their known bracketing relationship suggests they will typically give estimates with the same sign, although their tendency to over- or under-estimate the true effect will depend on how well the underlying assumptions of each are met (most notably, parallel trends for DiD and no unmeasured confounders for CA; see Introduction)[6,27]. Overall, these findings demonstrate the power of large within-study comparisons to directly quantify differences in the levels of bias associated with different designs.

We must acknowledge that the assumptions of our hierarchical model (that the bias for each design (j) is on average zero and normally distributed) cannot be verified without gold standard randomised experiments and that, for observational designs, the

model was overdispersed (potentially due to underestimation of statistical error by GLM(M)s or positively correlated design biases). The exact values of our hierarchical model should therefore be treated with appropriate caution, and future research is needed to refine and improve our approach to quantify these biases more precisely. Responses within datasets may also not be independent as multiple species could interact; therefore, the estimates analysed by our hierarchical model are statistically dependent on each other, and although we tried to account for this using a correlation matrix (see Methods, Eq. (3)), this is a limitation of our model. We must also recognise that we collated datasets using non-systematic searches[46,47] and therefore our analysis potentially exaggerates the intrinsic biases of observational designs (i.e., our data may disproportionately reflect situations where the BACI design was chosen to account for confounding factors).

**Table 2 Results of hierarchical Bayesian model for randomised and non-randomised datasets.**

| Term | Posterior mean | 95% Credible Interval |
|---|---|---|
| **Randomised (R-)** | | |
| $\sigma_\beta$ | 0.746 | [0.679, 0.813] |
| $\lambda$ | 1.119 | [0.980, 1.276] |
| $\sigma$[BACI DiD] | 0.029 | [0.005, 0.097] |
| $\sigma$[BACI CA] | 0.005 | [0.002, 0.008] |
| $\sigma$[CI] | 0.005 | [0.002, 0.008] |
| $\sigma$[BA] | 0.773 | [0.699, 0.846] |
| $\Omega$[BACI DiD, BACI CA] | 0.268 | [0.152, 0.379] |
| $\Omega$[BACI DiD, CI] | 0.239 | [0.122, 0.354] |
| $\Omega$[BACI DiD, BA] | 0.849 | [0.770, 0.914] |
| $\Omega$[BACI CA, CI] | 0.995 | [0.994, 0.996] |
| $\Omega$[BACI CA, BA] | −0.168 | [−0.332, 0.002] |
| $\Omega$[CI, BA] | −0.184 | [−0.349, −0.015] |
| **Non-randomised** | | |
| $\sigma_\beta$ | 0.700 | [0.628, 0.776] |
| $\lambda$ | 1.822 | [1.595, 2.098] |
| $\sigma$[BACI DiD] | 0.017 | [0.004, 0.049] |
| $\sigma$[BACI CA] | 0.049 | [0.005, 0.128] |
| $\sigma$[CI] | 0.091 | [0.008, 0.137] |
| $\sigma$[BA] | 0.645 | [0.573, 0.720] |
| $\Omega$[BACI DiD, BACI CA] | 0.140 | [0.010, 0.263] |
| $\Omega$[BACI DiD, CI] | 0.036 | [−0.106, 0.176] |
| $\Omega$[BACI DiD, BA] | 0.798 | [0.718, 0.865] |
| $\Omega$[BACI CA, CI] | 0.939 | [0.923, 0.954] |
| $\Omega$[BACI CA, BA] | −0.127 | [−0.285, 0.026] |
| $\Omega$[CI, BA] | −0.229 | [−0.397, −0.061] |

In randomised datasets, BACI and CI terms refer to R-BACI and R-CI designs (denoted by 'R-'). The $\sigma$ terms are the standard deviations of the bias of each design, so larger $\sigma$ values correspond to more biased designs. $\sigma_\beta$ refers to the standard deviation of the true effect across all datasets. $\Omega$ represents the within-response correlations between study design estimates, and $\lambda$ models systematic underestimation ($\lambda > 1$) or overestimation ($\lambda < 1$) of the statistical error using GLM (M)s. See methods for more details on the model.
BA before-after, BACI before-after-control-impact, CI control-impact.

We nevertheless show that researchers were wise to use the BACI design because it was less biased than CI and BA designs across a wide range of datasets from various environmental systems and locations. Without undertaking costly and time-consuming pre-impact sampling and pilot studies, researchers are also unlikely to know the levels of bias that could affect their results. Finally, we did not consider sample size, but it is likely that researchers might use larger sample sizes for CI and BA designs than BACI designs. This is, however, unlikely to affect our main conclusions because larger sample sizes could increase type I errors (false positive rate) by yielding more precise, but biased estimates of the true effect[28].

Our analyses provide several empirically supported recommendations for researchers designing future studies to assess an impact of interest. First, using a controlled and/or randomised design (if possible) was shown to strongly reduce the level of bias in study estimates. Second, when observational designs must be used (as randomisation is not feasible or too costly), we urge researchers to choose the BACI design over other observational designs—and when that is not possible, to choose the CI design over the uncontrolled BA design. We acknowledge that limited resources, short funding timescales, and ethical or logistical constraints[48] may force researchers to use the CI design (if randomisation and pre-impact sampling are impossible) or the BA design (if appropriate controls cannot be found[28]). To facilitate the usage of less biased designs, longer-term investments in research effort and funding are required[43]. Far greater emphasis on study designs in statistical education[49] and better training and collaboration between researchers, practitioners and methodologists, is needed to improve

the design of future studies; for example, potentially improving the CI design by pairing or matching the impact group and control group[22], or improving the BA design using regression discontinuity methods[48,50]. Where the choice of study design is limited, researchers must transparently communicate the limitations and uncertainty associated with their results.

Our findings also have wider implications for evidence synthesis, specifically the exclusion of certain observational study designs from syntheses (the 'rubbish in, rubbish out' concept[51,52]). We believe that observational designs should be included in systematic reviews and meta-analyses, but that careful adjustments are needed to account for their potential biases. Exclusion of observational studies often results from subjective, checklist-based 'Risk of Bias' or quality assessments of studies (e.g., AMSTRAD 2[53], ROBINS-I[54], or GRADE[55]) that are not data-driven and often neglect to identify the actual direction, or quantify the magnitude, of possible bias introduced by observational studies when rating the quality of a review's recommendations. We also found that there was a small proportion of studies that used randomised designs (R-CI or R-BACI) or observational BACI designs (Fig. 2), suggesting that systematic reviews and meta-analyses risk excluding a substantial proportion of the literature and limiting the scope of their recommendations if such exclusion criteria are used[32,56,57]. This problem is compounded by the fact that, at least in conservation science, studies using randomised or BACI designs are strongly concentrated in Europe, Australasia, and North America[31]. Systematic reviews that rely on these few types of study designs are therefore likely to fail to provide decision makers outside of these regions with locally relevant recommendations that they prefer[58]. The Covid-19 pandemic has highlighted the difficulties in making locally relevant evidence-based decisions using studies conducted in different countries with different demographics and cultures, and on patients of different ages, ethnicities, genetics, and underlying health issues[59]. This problem is also acute for decision-makers working on biodiversity conservation in the tropical regions, where the need for conservation is arguably the greatest (i.e., where most of Earth's biodiversity exists[60]) but they either have to rely on very few well-designed studies that are not locally relevant (i.e., have low generalisability), or more studies that are locally relevant but less well-designed[31,32]. Either option could lead decision-makers to take ineffective or inefficient decisions. In the long-term, improving the quality and coverage of scientific evidence and evidence syntheses across the world will help solve these issues, but shorter-term solutions to synthesising patchy evidence bases are required.

Our work furthers sorely needed research on how to combine evidence from studies that vary greatly in their design. Our approach is an alternative to conventional meta-analyses which tend to only weight studies by their sample size or the inverse of their variance[61]; when studies vary greatly in their study design, simply weighting by inverse variance or sample size is unlikely to account for different levels of bias introduced by different study designs (see Equation (1)). For example, a BA study could receive a larger weight if it had lower variance than a BACI study, despite our results suggesting a BA study usually suffers from greater design bias. Our model provides a principled way to weight studies by both their variance and the likely amount of bias introduced by their study design; it is therefore a form of 'bias-adjusted meta-analysis'[62–66]. However, instead of relying on elicitation of subjective expert opinions on the bias of each study, we provide a data-driven, empirical quantification of study biases – an important step that was called for to improve such meta-analytic approaches[65,66].

Future research is needed to refine our methodology, but our empirically grounded form of bias-adjusted meta-analysis could be implemented as follows: 1.) collate studies for the same true effect, their effect size estimates, standard errors, and the type of study design; 2.) enter these data into our hierarchical model,

where effect size estimates share the same intercept (the true causal effect), a random effect term due to design bias (whose variance is estimated by the method we used), and a random effect term for statistical noise (whose variance is estimated by the reported standard error of studies); 3.) fit this model and estimate the shared intercept/true effect. Heuristically, this can be thought of as weighting studies by both their design bias and their sampling variance and could be implemented on a dynamic meta-analysis platform (such as metadataset.com[67]). This approach has substantial potential to develop evidence synthesis in fields (such as biodiversity conservation[31,32]) with patchy evidence bases, where reliably synthesising findings from studies that vary greatly in their design is a fundamental and unavoidable challenge.

Our study has highlighted an often overlooked aspect of debates over scientific reproducibility: that the credibility of studies is fundamentally determined by study design. Testing the effectiveness of conservation and social interventions is undoubtedly of great importance given the current challenges facing biodiversity and society in general and the serious need for more evidence-based decision-making[1,68]. And yet our findings suggest that quantifiably less biased study designs are poorly represented in the environmental and social sciences. Greater methodological training of researchers and funding for intervention studies, as well as stronger collaborations between methodologists and practitioners is needed to facilitate the use of less biased study designs. Better communication and reporting of the uncertainty associated with different study designs is also needed, as well as more meta-research (the study of research itself) to improve standards of study design[69]. Our hierarchical model provides a principled way to combine studies using a variety of study designs that vary greatly in their risk of bias, enabling us to make more efficient use of patchy evidence bases. Ultimately, we hope that researchers and practitioners testing interventions will think carefully about the types of study designs they use, and we encourage the evidence synthesis community to embrace alternative methods for combining evidence from heterogeneous sets of studies to improve our ability to inform evidence-based decision-making in all disciplines.

## Methods

**Quantifying the use of different designs**. We compared the use of different study designs in the literature that quantitatively tested interventions between the fields of biodiversity conservation (4,260 studies collated by Conservation Evidence[45]) and social science (1,009 studies found by 32 systematic reviews produced by the Campbell Collaboration: www.campbellcollaboration.org).

Conservation Evidence is a database of intervention studies, each of which has quantitatively tested a conservation intervention (e.g., sowing strips of wildflower seeds on farmland to benefit birds), that is continuously being updated through comprehensive, manual searches of conservation journals for a wide range of fields in biodiversity conservation (e.g., amphibian, bird, peatland, and farmland conservation[45]). To obtain the proportion of studies that used each design from Conservation Evidence, we simply extracted the type of study design from each study in the database in 2019 – the study design was determined using a standardised set of criteria; reviews were not included (Table 3). We checked if the

designs reported in the database accurately reflected the designs in the original publication and found that for a random subset of 356 studies, 95.1% were accurately described.

Each systematic review produced by the Campbell Collaboration collates and analyses studies that test a specific social intervention; we collated systematic reviews that tested a variety of social interventions across several fields in the social sciences, including education, crime and justice, international development and social welfare (Supplementary Data 1). We retrieved systematic reviews produced by the Campbell Collaboration by searching their website (www.campbellcollaboration.org) for reviews published between 2013–2019 (as of 8th September 2019) — we limited the date range as we could not go through every review. As we were interested in the use of study designs in the wider social-science literature, we only considered reviews (32 in total) that contained sufficient information on the number of included and excluded studies that used different study designs. Studies may be excluded from systematic reviews for several reasons, such as their relevance to the scope of the review (e.g., testing a relevant intervention) and their study design. We only considered studies if the sole reason for their exclusion from the systematic review was their study design – i.e., reviews clearly reported that the study was excluded because it used a particular study design, and not because of any other reason, such as its relevance to the review's research questions. We calculated the proportion of studies that used each design in each systematic review (using the same criteria as for the biodiversity-conservation literature – see Table 3) and then averaged these proportions across all systematic reviews.

**Within-study comparisons of different study designs**. We wanted to make direct within-study comparisons between the estimates obtained by different study designs (e.g., see[38,70,71] for single within-study comparisons) for many different studies. If a dataset contains data collected using a BACI design, subsets of these data can be used to mimic the use of other study designs (a BA design using only data for the impact group, and a CI design using only data collected after the impact occurred). Similarly, if data were collected using a R-BACI design, subsets of these data can be used to mimic the use of a BA design and a R-CI design. Collecting BACI and R-BACI datasets would therefore allow us to make direct within-study comparisons of the estimates obtained by these designs.

We collated BACI and R-BACI datasets by searching the Web of Science Core Collection[72] which included the following citation indexes: Science Citation Index Expanded (SCI-EXPANDED) 1900-present; Social Sciences Citation Index (SSCI) 1900-present Arts & Humanities Citation Index (A&HCI) 1975-present; Conference Proceedings Citation Index - Science (CPCI-S) 1990-present; Conference Proceedings Citation Index - Social Science & Humanities (CPCI-SSH) 1990-present; Book Citation Index - Science (BKCI-S) 2008-present; Book Citation Index - Social Sciences & Humanities (BKCI-SSH) 2008-present; Emerging Sources Citation Index (ESCI) 2015-present; Current Chemical Reactions (CCR-EXPANDED) 1985-present (Includes Institut National de la Propriete Industrielle structure data back to 1840); Index Chemicus (IC) 1993-present. The following search terms were used: ['BACI'] OR ['Before-After Control-Impact'] and the search was conducted on the 18th December 2017. Our search returned 674 results, which we then refined by selecting only 'Article' as the document type and using only the following Web of Science Categories: 'Ecology', 'Marine Freshwater Biology', 'Biodiversity Conservation', 'Fisheries', 'Oceanography', 'Forestry', 'Zoology', 'Ornithology', 'Biology', 'Plant Sciences', 'Entomology', 'Remote Sensing', 'Toxicology' and 'Soil Science'. This left 579 results, which we then restricted to articles published since 2002 (15 years prior to search) to give us a realistic opportunity to obtain the raw datasets, thus reducing this number to 542. We were able to access the abstracts of 521 studies and excluded any that did not test the effect of an environmental intervention or threat using an R-BACI or BACI design with response measures related to the abundance (e.g., density, counts, biomass, cover), reproduction (reproductive success) or size (body length, body mass) of animals or plants. Many studies did not test a relevant metric (e.g., they measured species richness), did not use a BACI or R-BACI design, or did not test the effect of an intervention or threat — this left 96 studies for which we contacted all corresponding authors to ask for the raw dataset. We were able to fully access 54

---

**Table 3 Definitions used to categorise studies based on the study design they used.**

| Study design | Controlled? | Sampling before impact occurs? | Randomised allocation of replicates to the impact group and control group? |
|---|---|---|---|
| After | No | No | No |
| Before-after (BA) | No | Yes | No |
| Control-impact (CI) | Yes | No | No |
| Before-after control-impact (BACI) | Yes | Yes | No |
| Randomised control-impact (R-CI) | Yes | No | Yes |
| Randomised before-after control-impact (R-BACI) | Yes | Yes | Yes |

See also Fig. 1 for visual illustration and comparison of designs. Reviews from the database were not included.

raw datasets, but upon closer inspection we found that three of these datasets either: did not use a BACI design; did not use the metrics we specified; or did not provide sufficient data for our analyses. This left 51 datasets in total that we used in our preliminary analyses (Supplementary Data 2).

All the datasets were originally collected to evaluate the effect of an environmental intervention or impact. Most of them contained multiple response variables (e.g., different measures for different species, such as abundance or density for species A, B, and C). Within a dataset, we use the term "response" to refer to the estimation of the true effect of an impact on one response variable. There were 1,968 responses in total across 51 datasets. We then excluded 932 responses (resulting in the exclusion of one dataset) where one or more of the four time-period and treatment subsets (Before Control, Before Impact, After Control, and After Impact data) consisted of entirely zero measurements, or two or more of these subsets had more than 90% zero measurements. We also excluded one further dataset as it was the only one to not contain repeated measurements at sites in both the before- and after-periods. This was necessary to generate reliable standard errors when modelling these data. We modelled the remaining 1,036 responses from across 49 datasets (Supplementary Table 1).

We applied each study design to the appropriate components of each dataset using Generalised Linear Models (GLMs[73,74]) because of their generality and ability to implement the statistical estimators of many different study designs. The model structure of GLMs was adjusted for each response in each dataset based on the study design specified, response measure and dataset structure (Supplementary Table 2). We quantified the effect of the time period for the BA design (After vs Before the impact) and the effect of the treatment type for the CI and R-CI designs (Impact vs Control) on the response variable (Supplementary Table 2). For BACI and R-BACI designs, we implemented two statistical estimators: 1.) a DiD estimator that estimated the true effect using an interaction term between time and treatment type; and 2.) a covariance adjustment estimator that estimated the true effect using a term for the treatment type with a lagged variable (Supplementary Table 2).

As there were large numbers of responses, we used general a priori rules to specify models for each response; this may have led to some model misspecification, but was unlikely to have substantially affected our pairwise comparison of estimates obtained by different designs. The error family of each GLM was specified based on the nature of the measure used and preliminary data exploration: count measures (e.g., abundance) = poisson; density measures (e.g., biomass or abundance per unit area) = quasipoisson, as data for these measures tended to be overdispersed; percentage measures (e.g., percentage cover) = quasibinomial; and size measures (e.g., body length) = gaussian.

We treated each year or season in which data were collected as independent observations because the implementation of a seasonal term in models is likely to vary on a case-by-case basis; this will depend on the research questions posed by each study and was not feasible for us to consider given the large number of responses we were modelling. The log link function was used for all models to generate a standardised log response ratio as an estimate of the true effect for each response; a fixed effect coefficient (a variable named treatment status; Supplementary Table 2) was used to estimate the log response ratio[61]. If the response had at least ten 'sites' (independent sampling units) and two measurements per site on average, we used the random effects of subsample (replicates within a site) nested within site to capture the dependence within a site and subsample (i.e., a Generalised Linear Mixed Model or GLMM[73,74] was implemented instead of a GLM); otherwise we fitted a GLM with only the fixed effects (Supplementary Table 2).

We fitted all models using R version 3.5.1[75], and packages lme4[76] and MASS[77]. Code to replicate all analyses is available (see Data and Code Availability). We compared the estimates obtained using each study design (both in terms of point estimates and estimates with associated standard error) by their magnitude and sign.

**A model-based quantification of the bias in study design estimates**. We used a hierarchical Bayesian model motivated by the decomposition in Equation (1) to quantify the bias in different study design estimates. This model takes the estimated effects of impacts and their standard errors as inputs. Let $\hat{\beta}_{ij}$ be the true effect estimator in study $i$ using design $j$ and $\hat{\sigma}_{ij}$ be its estimated standard error from the corresponding GLM or GLMM. Our hierarchical model assumes:

$$\hat{\beta}_{ij} = \beta_i + \gamma_{ij} + \varepsilon_{ij},$$
$$\beta_i \sim N(0, \sigma_\beta^2), \quad \gamma_{ij} \sim N(0, \sigma_j^2), \quad \varepsilon_i \sim N(0, \Lambda), \quad (2)$$

where $\beta_i$ is the true effect for response $i$, $\gamma_{ij}$ is the bias of design $j$ in response $i$, and $\varepsilon_{ij}$ is the sampling noise of the statistical estimator. Although $\gamma_{ij}$ technically incorporates both the design bias and any misspecification (modelling) bias due to using GLMs or GLMMs (Equation (1)), we expect the modelling bias to be much smaller than the design bias[3,11]. We assume the statistical errors $\varepsilon_i$ within a response are related to the estimated standard errors through the following joint distribution:

$$\Lambda = \lambda \cdot \text{diag}(\hat{\sigma}_i)\Omega\text{diag}(\hat{\sigma}_i), \quad (3)$$

where $\Omega$ is the correlation matrix for the different estimators in the same response and $\lambda$ is a scaling factor to account for possible over/under-estimation of the standard errors.

This model effectively quantifies the bias of design $j$ using the value of $\sigma_j$ (larger values = more bias) by accounting for within-response correlations using the correlation matrix $\Omega$ and for possible under-estimation of the standard error using $\lambda$. We ensured that the prior distributions we used had very large variances so they would have a very small effect on the posterior distribution — accordingly we placed the following disperse priors on the variance parameters:

$$\sigma_\beta, \sigma_1, \dots, \sigma_J \sim \text{Inv-Gamma}(1, 0.02), \quad \lambda \sim \text{Gamma}(2, 2), \quad \Omega \sim \text{LKJ}(1) \quad (4)$$

We fitted the hierarchical Bayesian model in R version 3.5.1 using the Bayesian inference package rstan[78].

## Data availability

All data analysed in the current study are available from Zenodo, https://doi.org/10.5281/zenodo.3560856. Source data are provided with this paper.

## Code availability

All code used in the current study is available from Zenodo, https://doi.org/10.5281/zenodo.3560856.

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

## Acknowledgements

We are grateful to the following people and organisations for contributing datasets to this analysis: P. Edwards, G.R. Hodgson, H. Welsh, J.V. Vieira, authors of van Deurs et al. 2012, T. M. Grome, M. Kaspersen, H. Jensen, C. Stenberg, T. K. Sørensen, J. Støttrup, T. Warnar, H. Mosegaard, Axel Schwerk, Alberto Velando, Dolores River Restoration Partnership, J.S. Pinilla, A. Page, M. Dasey, D. Maguire, J. Barlow, J. Louzada, Jari Florestal, R.T. Buxton, C.R. Schacter, J. Seoane, M.G. Conners, K. Nickel, G. Marakovich, A. Wright, G. Soprone, CSIRO, A. Elosegi, L. García-Arberas, J. Díez, A. Rallo, Parks and Wildlife Finland, Parc Marin de la Côte Bleue. Author funding sources: T.A. was supported by the Grantham Foundation for the Protection of the Environment, Kenneth Miller Trust and Australian Research Council Future Fellowship (FT180100354); W.J.S. and P.A.M. were supported by Arcadia, MAVA, and The David and Claudia Harding Foundation; A.P.C. was supported by the Natural Environment Research Council via

Cambridge Earth System Science NERC DTP (NE/L002507/1); D.A. was funded by Portugal national funds through the FCT – Foundation for Science and Technology, under the Transitional Standard – DL57 / 2016 and through the strategic project UIDB/04326/2020; M.A. acknowledges Koniambo Nickel SAS, and particularly Gregory Marakovich and Andy Wright; J.C.A. was funded through by Dirección General de Investigación Científica, projects PB97-1252, BOS2002-01543, CGL2005-04893/BOS, CGL2008-02567 and Comunidad de Madrid, as well as by contract HENARSA-CSIC 2003469-CSIC19637; A.A. was funded by Spanish Government: MEC (CGL2007-65176); B.P.B. was funded through the U.S. Geological Survey and the New York City Department of Environmental Protection; R.B. was funded by Comunidad de Madrid (2018-T1/AMB-10374); J.A.S. and D.A.B. were funded through the U.S. Geological Survey and NextEra Energy; R.S.C. was funded by the Portuguese Foundation for Science and Technology (FCT) grant SFRH/BD/78813/2011 and strategic project UID/MAR/04292/2013; A.D.B. was funded through the Belgian offshore wind monitoring program (WINMON-BE), financed by the Belgian offshore wind energy sector via RBINS—OD Nature; M.K.D. was funded by the Harold L. Castle Foundation; P.M.E. was funded by the Clackamas County Water Environment Services River Health Stewardship Program and the Portland State University Student Watershed Research Project; T.D.E., J.P.A.G. and A.P. were supported by funding from the New Zealand Department of Conservation (Te Papa Atawhai) and from the Centre for Marine Environmental & Economic Research, Victoria University of Wellington, New Zealand; F.M.F. was funded by CNPq-CAPES grants (PELD site 23 403811/2012-0, PELD-RAS 441659/2016-0, BEX5528/13-5 and 383744/2015-6) and BNP Paribas Foundation (Climate & Biodiversity Initiative, BIOCLIMATE project); B.P.H. was funded by NOAA-NMFS sea scallop research set-aside program awards NA16FM1031, NA06FM1001, NA16FM2416, and NA04NMF4720332; A.L.B. was funded by the Portuguese Foundation for Science and Technology (FCT) grant FCT PD/BD/52597/2014, Bat Conservation International student research fellowship and CNPq grant 160049/2013-0; L.C.M. acknowledges Secretaría de Ciencia y Técnica (UNRC); R.A.M. acknowledges Alaska Fisheries Science Center, NOAA Fisheries, and U.S. Department of Commerce for salary support; C.F.J.M. was funded by the Portuguese Foundation for Science and Technology (FCT) grant SFRH/BD/80488/2011; R.R. was funded by the Portuguese Foundation for Science and Technology (FCT) grant PTDC/BIA-BIC/111184/2009, by Madeira's Regional Agency for the Development of Research, Technology and Innovation (ARDITI) grant M1420-09-5369-FSE-000002 and by a Bat Conservation International student research fellowship; J.C. and S.S. were funded by the Alabama Department of Conservation and Natural Resources; A.T. was funded by the Spanish Ministry of Education with a Formacion de Profesorado Universitario (FPU) grant AP2008-00577 and Dirección General de Investigación Científica, project CGL2008-02567; C.W. was funded by Strategic Science Investment Funding of the Ministry of Business, Innovation and Employment, New Zealand; J.S.K. acknowledges Boreal Peatland LIFE (LIFE08 NAT/FIN/000596), Parks and Wildlife Finland and Kone Foundation; J.J.S.S. was funded by the Mexican National Council on Science and Technology (CONACYT 242558); N.N. was funded by The Carl Tryggers Foundation; I.L.J. was funded by a Discovery Grant from the Natural Sciences and Engineering Research Council of Canada; D.D. and D.S. were funded by the French National Research Agency via the "Investment for the Future" program IDEALG (ANR-10-BTBR-04) and by the ALGMARBIO project; R.C.P. was funded by CSIRO and whose research was also supported by funds from the Great Barrier Reef Marine Park Authority, the Fisheries Research and Development Corporation, the Australian Fisheries Management Authority, and Queensland Department of Primary Industries (QDPI). Any use of trade, firm, or product names is for descriptive purposes only and does not imply endorsement by the U.S. Government. The scientific results and conclusions, as well as any views or opinions expressed herein, are those of the author(s) and do not necessarily reflect those of NOAA or the Department of Commerce.

## Author contributions

A.P.C., T.A., P.A.M., Q.Z., and W.J.S. designed the research; A.P.C. wrote the paper; D.A., M.A., J.C.A., A.A., B.P.B, R.B., J.B., D.A.B., J.C., R.S.C., L.C.M., S.C., J.C., M.D.C, D.D., A.D.B., M.K.D., T.D.E., P.M.E., F.M.F., J.P.A.G., B.P.H., A.H., I.L.J., B.P.K., J.S.K., A.L.B., H.L.M., A.M., B.M., C.A.M., D.M., R.A.M, M.M., C.F.J.M.,K.M., M.M., N.N., C.P., A.P., C.R.P., C.P., M.R., R.R., M.C.R., J.J.S.S., J.A.S., S.S., A.A.S., D.S., K.D.E.S., T.R.S., A.T., O.T., T.V., C.W. contributed datasets for analyses. All authors reviewed, edited, and approved the manuscript.

## Competing interests

The authors declare no competing interests.

## Additional information

Alec P. Christie [1✉], David Abecasis [2], Mehdi Adjeroud[3], Juan C. Alonso [4], Tatsuya Amano [5], Alvaro Anton [6], Barry P. Baldigo [7], Rafael Barrientos [8], Jake E. Bicknell [9], Deborah A. Buhl[10], Just Cebrian [11], Ricardo S. Ceia [12,13], Luciana Cibils-Martina [14,15], Sarah Clarke[16], Joachim Claudet [17], Michael D. Craig[18,19], Dominique Davoult[20], Annelies De Backer [21], Mary K. Donovan [22,23], Tyler D. Eddy[24,25,26], Filipe M. França [27], Jonathan P. A. Gardner [26], Bradley P. Harris[28], Ari Huusko[29], Ian L. Jones[30], Brendan P. Kelaher[31], Janne S. Kotiaho [32,33], Adrià López-Baucells [34,35,36], Heather L. Major [37], Aki Mäki-Petäys[38,39], Beatriz Martín[40,41], Carlos A. Martín[8], Philip A. Martin[1,42], Daniel Mateos-Molina [43], Robert A. McConnaughey [44], Michele Meroni[45], Christoph F. J. Meyer [34,35,46], Kade Mills[47], Monica Montefalcone[48], Norbertas Noreika [49,50], Carlos Palacín[4], Anjali Pande[26,51,52], C. Roland Pitcher [53], Carlos Ponce[54], Matt Rinella[55], Ricardo Rocha [34,35,56], María C. Ruiz-Delgado[57], Juan J. Schmitter-Soto [58], Jill A. Shaffer [10], Shailesh Sharma [59], Anna A. Sher [60], Doriane Stagnol[20],

Thomas R. Stanley[61], Kevin D. E. Stokesbury[62], Aurora Torres[63,64], Oliver Tully[16], Teppo Vehanen [65], Corinne Watts[66], Qingyuan Zhao[67] & William J. Sutherland[1,42]

[1]Conservation Science Group, Department of Zoology, University of Cambridge, The David Attenborough Building, Downing Street, Cambridge CB3 3QZ, UK. [2]Centre of Marine Sciences (CCMar), Universidade do Algarve, Campus de Gambelas 8005-139 Faro, Portugal. [3]Institut de Recherche pour le Développement (IRD), UMR 9220 ENTROPIE & Laboratoire d'Excellence CORAIL, Université de Perpignan Via Domitia, 52 avenue Paul Alduy, 66860 Perpignan, France. [4]Museo Nacional de Ciencias Naturales, CSIC, Madrid, Spain. [5]School of Biological Sciences, University of Queensland, Brisbane 4072 QLD, Australia. [6]Education Faculty of Bilbao, University of the Basque Country (UPV/EHU). Sarriena z/g E-48940 Leioa, Basque Country, Spain. [7]U.S. Geological Survey, New York Water Science Center, 425 Jordan Rd., Troy, NY 12180, USA. [8]Universidad Complutense de Madrid, Departamento de Biodiversidad, Ecología y Evolución, Facultad de Ciencias Biológicas, c/ José Antonio Novais, 12, E-28040 Madrid, Spain. [9]Durrell Institute of Conservation and Ecology (DICE), School of Anthropology and Conservation, University of Kent, Canterbury CT2 7NR, UK. [10]U.S. Geological Survey, Northern Prairie Wildlife Research Center, Jamestown, ND 58401, USA. [11]Northern Gulf Institute, Mississippi State University, 1021 Balch Blvd, John C. Stennis Space Center, Mississippi 39529, USA. [12]MARE – Marine and Environmental Sciences Centre, Dept. Life Sciences, University of Coimbra, Coimbra, Portugal. [13]CFE – Centre for Functional Ecology, Dept. Life Sciences, University of Coimbra, Coimbra, Portugal. [14]Departamento de Ciencias Naturales, Universidad Nacional de Río Cuarto (UNRC), Córdoba, Argentina. [15]CONICET, Buenos Aires, Argentina. [16]Marine Institute, Rinville, Oranmore, Galway, Ireland. [17]National Center for Scientific Research, PSL Université Paris, CRIOBE, USR 3278 CNRS-EPHE-UPVD, Maison des Océans, 195 rue Saint-Jacques, 75005 Paris, France. [18]School of Biological Sciences, University of Western Australia, Nedlands, WA 6009, Australia. [19]School of Environmental and Conservation Sciences, Murdoch University, Murdoch, WA 6150, Australia. [20]Sorbonne Université, CNRS, UMR 7144, Station Biologique, F.29680 Roscoff, France. [21]Flanders Research Institute for Agriculture, Fisheries and Food (ILVO), Ankerstraat 1, 8400 Ostend, Belgium. [22]Marine Science Institute, University of California Santa Barbara, Santa Barbara, CA 93106, USA. [23]Hawaii Institute of Marine Biology, University of Hawaii at Manoa, Honolulu, HI 96822, USA. [24]Baruch Institute for Marine & Coastal Sciences, University of South Carolina, Columbia, SC, USA. [25]Centre for Fisheries Ecosystems Research, Fisheries & Marine Institute, Memorial University of Newfoundland, St. John's, Canada. [26]School of Biological Sciences, Victoria University of Wellington, P O Box 600, Wellington 6140, New Zealand. [27]Lancaster Environment Centre, Lancaster University, LA1 4YQ Lancaster, UK. [28]Fisheries, Aquatic Science and Technology Laboratory, Alaska Pacific University, 4101 University Dr., Anchorage, AK 99508, USA. [29]Natural Resources Institute Finland, Manamansalontie 90, 88300 Paltamo, Finland. [30]Department of Biology, Memorial University, St. John's, NL A1B 2R3, Canada. [31]National Marine Science Centre and Marine Ecology Research Centre, Southern Cross University, 2 Bay Drive, Coffs Harbour 2450, Australia. [32]Department of Biological and Environmental Science, University of Jyväskylä, Jyväskylä, Finland. [33]School of Resource Wisdom, University of Jyväskylä, Jyväskylä, Finland. [34]Centre for Ecology, Evolution and Environmental Changes – cE3c, Faculty of Sciences, University of Lisbon, 1749-016 Lisbon, Portugal. [35]Biological Dynamics of Forest Fragments Project, National Institute for Amazonian Research and Smithsonian Tropical Research Institute, 69011-970 Manaus, Brazil. [36]Granollers Museum of Natural History, Granollers, Spain. [37]Department of Biological Sciences, University of New Brunswick, PO Box 5050, Saint John, NB E2L 4L5, Canada. [38]Voimalohi Oy, Voimatie 23, Voimatie 91100 Ii, Finland. [39]Natural Resources Institute Finland, Paavo Havaksen tie 3, 90014 University of Oulu, Oulu, Finland. [40]Fundación Migres CIMA Ctra, Cádiz, Spain. [41]Intergovernmental Oceanographic Commission of UNESCO, Marine Policy and Regional Coordination Section Paris 07, Paris, France. [42]BioRISC, St. Catharine's College, Cambridge CB2 1RL, UK. [43]Departamento de Ecología e Hidrología, Universidad de Murcia, Campus de Espinardo 30100 Murcia, Spain. [44]RACE Division, Alaska Fisheries Science Center, National Marine Fisheries Service, NOAA, 7600 Sand Point Way NE, Seattle, WA 98115, USA. [45]European Commission, Joint Research Centre (JRC), Ispra, VA, Italy. [46]School of Science, Engineering and Environment, University of Salford, Salford M5 4WT, UK. [47]Victorian National Park Association, Carlton, VIC, Australia. [48]Department of Earth, Environment and Life Sciences (DiSTAV), University of Genoa, Corso Europa 26, 16132 Genoa, Italy. [49]Department of Ecology, Swedish University of Agricultural Sciences, Uppsala, Sweden. [50]Chair of Plant Health, Institute of Agricultural and Environmental Sciences, Estonian University of Life Sciences, Tartu, Estonia. [51]Biosecurity New Zealand – Tiakitanga Pūtaiao Aotearoa, Ministry for Primary Industries – Manatū Ahu Matua, 66 Ward St, PO Box 40742, Wallaceville, New Zealand. [52]National Institute of Water & Atmospheric Research Ltd (NIWA), 301 Evans Bay Parade, Greta Point Wellington, New Zealand. [53]CSIRO Oceans & Atmosphere, Queensland Biosciences Precinct, 306 Carmody Road, ST. LUCIA QLD 4067, Australia. [54]Museo Nacional de Ciencias Naturales, CSIC, José Gutiérrez Abascal 2, E-28006 Madrid, Spain. [55]Fort Keogh Livestock and Range Research Laboratory, 243 Fort Keogh Rd, Miles City, Montana 59301, USA. [56]CIBIO-InBIO, Research Centre in Biodiversity and Genetic Resources, University of Porto, Vairão, Portugal. [57]Departamento de Sistemas Físicos, Químicos y Naturales, Universidad Pablo de Olavide, ES-41013 Sevilla, Spain. [58]El Colegio de la Frontera Sur, A.P. 424, 77000 Chetumal, QR, Mexico. [59]Division of Fish and Wildlife, New York State Department of Environmental Conservation, 625 Broadway, Albany, NY 12233-4756, USA. [60]University of Denver Department of Biological Sciences, Denver, CO, USA. [61]U.S. Geological Survey, Fort Collins Science Center, Fort Collins, CO 80526, USA. [62]School for Marine Science and Technology, University of Massachusetts Dartmouth, New Bedford, MA, USA. [63]Georges Lemaître Earth and Climate Research Centre, Earth and Life Institute, Université Catholique de Louvain, 1348 Louvain-la-Neuve, Belgium. [64]Center for Systems Integration and Sustainability, Department of Fisheries and Wildlife, 13 Michigan State University, East Lansing, MI 48823, USA. [65]Natural Resources Institute Finland, Latokartanonkaari 9, 00790 Helsinki, Finland. [66]Manaaki Whenua – Landcare Research, Private Bag 3127, Hamilton 3216, New Zealand. [67]Statistical Laboratory, Department of Pure Mathematics and Mathematical Statistics, University of Cambridge, Wilberforce Road, Cambridge CB3 0WB, UK. ✉email: alec.christie@hotmail.co.uk

