## [Peer Review File · Nature Communications]

Reviewers' comments, first round:

Reviewer #1 (Remarks to the Author):

This project reviews study designs that are used in measuring environmental impacts, and tries to estimate the effect of using a quasi-experimental design in an original meta-analysis. This is an interesting idea, and a lot of work has gone into this paper, but there are some important flaws limiting its accessibility.

To look at the effect of design on results, datasets analysed using more complete designs (BACI) were broken into subsets (e.g. ignoring control group or pre-impact data) to study the cost of using an incomplete design. A problem with this approach is selection bias – by only using studies which used a BACI design, it is hard to make inferences beyond study systems for which BACI might be used. e.g. a researcher may be more likely to use a BACI design if concerned about pre-impact confounders, hence the effect they have on results might be exaggerated here. Also, is more care taken in choice of controls in control-impact studies that lack pre-impact measurements (because of awareness of the inferential risks)? These questions cannot be answered from the current study. Essentially, this meta-analysis is essentially a quasi-experimental before-after study of BACI designs! Its limitations should be made more explicit.

A larger issue was the nature of the analyses themselves. The analyses that were actually done were quite confusing and not well explained in the main manuscript, more intuition into what was done and why is needed. In addition, some specific issues:

- A lot of metrics were constructed which were a long way from the original data, and there was little insight available into actual results from the different studies, and importantly, how they might have actually changed under different designs. One way forward is through better visualisations, e.g. you could use a scatterplot with BACI up one axis and CI (or BA) on the other with a point for each study that represents its effect size (ideally, with some confidence bounds). If these points move far from the 1:1 line (how far actually matters, 20%? 40%?) this is an indication that results would have changed qualitatively by changing design.
- Random subsampling was done in analyses, the reason for doing so was not apparent. This seems to be a distraction that overly complicates analysis, a straight comparison of full datasets more directly answers the question of interest.
- One of the metrics used was "noncoverage probability", looking at how often the CI for effect size from a random subsample captured the value from BACI of the full dataset. A problem with this is that the value from the full dataset is itself random and it is not obvious that it should fall inside any of the other confidence intervals (if treating the observed dataset as the population and subsampling from it, an exercise of questionable relevance, one would need to make finite population corrections). Another issue is that this metric measures statistical significance not scientific significance – you could have a non-coverage prob of 100%, which sounds bad, but if all metrics are missing by about 2% it's not such a problem!

Minor comments:

This project has a very large interdisciplinary team, but it is unfortunate that it lacks a statistician. The focus of the paper is on statistical practice, especially in relation to study design and implications for meta-analysis.

The word "robust" is used a lot in the paper and in a different way to how it is used statistically – robustness usually refers to ability to handle extreme values in data, but it is used here to refer to BACI designs less likely to change qualitatively in results if there is no control or no pre-impact measurements. A better way to say this might be to talk about sensitivity to study design – how sensitive are results to choice of study design (borrowing language from the modelling field, which is big on sensitivity analysis).

Lines 201-2: strictly speaking they remove confounding, they don't minimise it, and this is not a

function of sample size. The authors seem to be confusing any given realisation of randomisation with the randomisation process itself (which allows pre-impact "confounders" to be treated like sampling error). You don't need a large sample size to remove a confounder – a properly designed experiment removes it at any sample size. Sample size is important for a different reason – to see patterns in the presence of sampling variation (which arises, amongst other reasons, from the pre-impact "confounders")

Line 209 what is meant by "true randomisation"? Does this just mean randomisation (as opposed to haphazard sampling)?

Figure 1 is a nice illustration of the designs being compared.

Line 393: "severely compromise" – where was this shown? Comments here need to be substantiated by results

Lines 410-412: but how would you weight by study design? Without a concrete recommendation this cannot be implemented. An alternative and more statistically sound approach would be to treat the missing data as missing data, e.g. use multiple imputation to impute pre-impact values when missing. Multiple imputation is already becoming widely used in metaanalysis in ecology, e.g.

<https://besjournals.onlinelibrary.wiley.com/doi/full/10.1111/2041-210X.12322>

Reviewer #2 (Remarks to the Author):

In this manuscript, the long list of authors provide meta-information on over six thousand studies from two scientific fields. This must have been a very labourous exercise and I think the results presented are certainly of interest to a wider scientific audience.

One major thing which seems lacking is an explanation of why scientists might use designs that are inferior to Randomized Controlled Trials (RCTs). RCTs are indeed the gold standard, but not always feasible due to financial, ethical or other constraints. For a short reflection on this, see Section 3.1 and -especially- the references therein of <http://dx.doi.org/10.1016/j.invent.2018.02.002>. In the discussion section the authors seem to imply that all scientific studies should be RCTs (or other gold standard methods), yet this simply is not feasible - not even within experimental (i.e., non-observational) designs.

In line with this, I find the "if the design isn't extremely robust, it's hurting science"-narrative too harsh (see e.g. the opening lines of the abstract and the discussion section). In my view, there's nothing wrong with performing studies with less robust designs, just as it is perfectly valued to perform a small-sample study, *as long as* you don't overclaim your findings. If you only have an "After"-design, or if you have a sample that is too small and/or too homogeneous or a convenience sample, there are still conclusions that one could draw from the study; however these, obviously, are much more limited than those that can be drawn from large scale RCTs.

It might be a tedious task, but it would certainly add value to the manuscript if the authors study to what extend the authors of the 6,532 papers studied indeed overclaim their results. If the authors don't overclaim on a massive scale, the harm for science of such 'inferior designs' is more limited and the paper should then be toned down accordingly. (Note: also in this situation the results are certainly publishable in my view.)

In the discussion/implication section, I would make a subsection with implications for meta research. I agree with the authors that for meta analyses, the type of research design certainly matters and that one cannot simply average the results of a RCT with those of some observational study. However, as outlined in the previous, I'm reluctant to go as far as the authors with the implications for science in general.

Finally, I want to thank the authors for the extensive amount of 'supplementary materials'. They did a good job of separating the main message (which is in the manuscript) from details that are only of interest to a small proportion of readers (such as meta-analysts; these materials are the supplementary materials). Due to time limitations on my side, I did not thoroughly review the supplementary materials.

Signed,
Casper Albers

Reviewer #3 (Remarks to the Author):

This paper presents an analysis of the prevalence of the use of various kinds of experimental designs in social science and environmental research. While the analyses are fine, the interpretation of the results has limitations.

I take issue with the claim that they have demonstrated that the designs have "severely compromised the results" (line 393). They quantified the deviations. But they did not indicate how often the overall conclusion of the study was changed. We often are not interested in the precise parameter estimate, but rather the general direction of the effect.

While mention is made of meta-analysis, the authors do not grapple with the question of whether these less-than-perfect designs, while problematic as single studies, still provide robust information when combined into a meta-analysis. If the main problem with these designs is simply an increase in the variance of the results, then the mean effect may still be robust. That is not true if the results are biased, but nowhere in this paper do the authors directly address that issue, except by anecdotal mentions of particular changes.

I am also leery of the suggestion of weighting evidence in a meta-analysis by the study design. What sort of weighting factor would you use? Do you weigh a BACI study as twice the value of a Before-After study? Three times? It seems impossible to know how to weigh them without knowing the "true" effect.

Nowhere do the authors discuss some of the reasons that a less than ideal design is used. For both conservation and social questions, experimental manipulations may be unethical or logistically unfeasible (e.g., diversity patterns over landscapes). Does that mean that we can never study those questions? No. It just means that in any such study the limitations of the data need to be acknowledged.

One of the main points of the paper seems to be that a bigger sample size is better. This is hardly news. Importantly, the effects are all in terms of a percentage decrease of the original sample size. But a decrease of 50% when you start from 200 is not the same as when you start from 20, as statistical power is not linear with sample size. On the other hand, the authors make a point about the percentage of randomized control trials that had a different sign (line 320) based on a sample size of 5.

Title: As the number of such designs in other arenas is not quantified, the word "more" should be deleted from the title.

Reviewer #1:**Comment:**

This project reviews study designs that are used in measuring environmental impacts, and tries to estimate the effect of using a quasi-experimental design in an original meta-analysis. This is an interesting idea, and a lot of work has gone into this paper, but there are some important flaws limiting its accessibility.

To look at the effect of design on results, datasets analysed using more complete designs (BACI) were broken into subsets (e.g. ignoring control group or pre-impact data) to study the cost of using an incomplete design. A problem with this approach is selection bias – by only using studies which used a BACI design, it is hard to make inferences beyond study systems for which BACI might be used. e.g. a researcher may be more likely to use a BACI design if concerned about pre-impact confounders, hence the effect they have on results might be exaggerated here. Also, is more care taken in choice of controls in control-impact studies that lack pre-impact measurements (because of awareness of the inferential risks)? These questions cannot be answered from the current study. Essentially, this meta-analysis is essentially a quasi-experimental before-after study of BACI designs! Its limitations should be made more explicit.

Response:

We agree that this a potential source of bias that must be discussed. We are effectively estimating the (causal) reduction of bias of using a BACI design, among those studies that the researchers chose to use a BACI design. We dedicate some text in the discussion to transparently addressing this issue and other limitations (L399-417), but we do not believe this possible bias is likely to change the main qualitative conclusions of our manuscript. Our results show that the choice of these researchers was a wise one as in those situations BACI indeed appeared to be more reliable than the other designs. There is also no way of a researcher knowing whether pre-confounding factors are likely to exist, or their degree of magnitude, prior to conducting a study, so the precautionary principle should be applied to avoid such biases.

Comment:

A larger issue was the nature of the analyses themselves. The analyses that were actually done were quite confusing and not well explained in the main manuscript, more intuition into what was done and why is needed.

Response:

We have taken care to more carefully explain our new analyses which are easier to explain than those in the previous version of the manuscript (L550-632).

Comment:

In addition, some specific issues:

- A lot of metrics were constructed which were a long way from the original data, and there was little insight available into actual results from the different studies, and importantly, how they might have actually changed under different designs. One way forward is through better visualisations, e.g. you could use a scatterplot with BACI up one axis and CI (or BA) on the other with a point for each study that represents its effect size (ideally, with some confidence bounds). If these points move far from

the 1:1 line (how far actually matters, 20%? 40%?) this is an indication that results would have changed qualitatively by changing design.

Response:

Our new analyses use estimates obtained from GLMMs and present our data in a completely different way (L550-632). We took this advice onboard and present a similar Figure (Fig.3) to what the reviewer describes here, making more direct pairwise, within-study comparisons of the estimates given by different designs, as well as metrics that are closer to the original data.

Comment:

- Random subsampling was done in analyses, the reason for doing so was not apparent. This seems to be a distraction that overly complicates analysis, a straight comparison of full datasets more directly answers the question of interest.

Response:

Our new analyses are not overcomplicated and do not involve this random subsampling to compare the effect of sample sizes. We agree that our new analyses now directly answers the question of interest better.

Comment:

- One of the metrics used was “noncoverage probability”, looking at how often the CI for effect size from a random subsample captured the value from BACI of the full dataset. A problem with this is that the value from the full dataset is itself random and it is not obvious that it should fall inside any of the other confidence intervals (if treating the observed dataset as the population and subsampling from it, an exercise of questionable relevance, one would need to make finite population corrections). Another issue is that this metric measures statistical significance not scientific significance – you could have a non-coverage prob of 100%, which sounds bad, but if all metrics are missing by about 2% it’s not such a problem!

Response:

Our new analyses do not focus as much on these metrics now, in favour of a more novel comparison of the reliability of the different designs using a Bayesian hierarchical model that compares the variance in the bias of different designs (L634-663). Where we do use certain metrics to undertake a pairwise comparison of the estimates of different designs, we compare point estimates by their magnitude and sign, and the associated 95% confidence intervals in terms of whether both designs give statistically significant results, whether their confidence intervals overlap, and whether they both lie entirely above or below zero. These metrics are more closely tied to the data and directly compare how similar two estimates given by different designs are for each pairwise comparison across all datasets. We agree that the non-coverage probability was not a sensible measure to use when using random subsampling, and should also be used in conjunction with other measures that look at scientific significance (e.g. magnitude and sign).

Comment:

Minor comments:

This project has a very large interdisciplinary team, but it is unfortunate that it lacks a statistician.

The focus of the paper is on statistical practice, especially in relation to study design and implications for meta-analysis.

Response:

We agree which is why we included a new co-author – Qingyuan Zhao – who is a statistician with expertise in causal inference and study design. He has substantially improved the credibility of the manuscript with new analyses. He has also helped us to come up with a new methodology to quantify the reliability of different study designs, which now forms a major part of our new manuscript. We have also consulted more extensively with Deb Buhl who is also an ecological statistician in our team.

Comment:

The word “robust” is used a lot in the paper and in a different way to how it is used statistically – robustness usually refers to ability to handle extreme values in data, but it is used here to refer to BACI designs less likely to change qualitatively in results if there is no control or no pre-impact measurements. A better way to say this might be to talk about sensitivity to study design – how sensitive are results to choice of study design (borrowing language from the modelling field, which is big on sensitivity analysis).

Response:

We have removed the word ‘robust’ in favour of speaking in terms of bias throughout, as we believe this is more appropriate and easier to understand for the broad audience we are trying to appeal to.

Comment:

Lines 201-2: strictly speaking they remove confounding, they don’t minimise it, and this is not a function of sample size. The authors seem to be confusing any given realisation of randomisation with the randomisation process itself (which allows pre-impact “confounders” to be treated like sampling error). You don’t need a large sample size to remove a confounder – a properly designed experiment removes it at any sample size. Sample size is important for a different reason – to see patterns in the presence of sampling variation (which arises, amongst other reasons, from the pre-impact “confounders”)

Response:

We agree this was an error and have clarified this when explaining the differences between study designs in the manuscript (L230-232).

Comment:

Line 209 what is meant by “true randomisation”? Does this just mean randomisation (as opposed to haphazard sampling)?

Response:

This is no longer mentioned in the updated manuscript.

Comment:

Figure 1 is a nice illustration of the designs being compared.

Response:

We are glad the reviewer liked this figure – we have expanded this to better illustrate the different study designs we are considering.

Comment:

Line 393: “severely compromise” – where was this shown? Comments here need to be substantiated by results

Response:

We agree this was overstating our results and this is no longer present in the updated manuscript. We have rewritten, rephrased and reframed the paper to address concerns by all reviewer that we were advocating against using these less-than-ideal designs. On the contrary, we were arguing the opposite and it was not helpful to use language like this as it clearly distorted our true messages from being effectively communicated.

Comment:

Lines 410-412: but how would you weight by study design? Without a concrete recommendation this cannot be implemented. An alternative and more statistically sound approach would be to treat the missing data as missing data, e.g. use multiple imputation to impute pre-impact values when missing. Multiple imputation is already becoming widely used in metaanalysis in ecology, e.g.

<https://besjournals.onlinelibrary.wiley.com/doi/full/10.1111/2041-210X.12322>

Response:

Using our new analyses we present a new methodology to do exactly this. We present a hierarchical model that provides a principled approach to combining studies that vary greatly in their design based on our quantification of the reliability of each design (L634-663).

Reviewer #2:

Comment:

In this manuscript, the long list of authors provide meta-information on over six thousand studies from two scientific fields. This must have been a very labourous exercise and I think the results presented are certainly of interest to a wider scientific audience.

One major thing which seems lacking is an explanation of why scientists might use designs that are inferior to Randomized Controlled Trials (RCTs). RCTs are indeed the gold standard, but not always feasible due to financial, ethical or other constraints. For a short reflection on this, see Section 3.1 and -especially- the references therein of <http://dx.doi.org/10.1016/j.invent.2018.02.002>. In the discussion section the authors seem to imply that all scientific studies should be RCTs (or other gold standard methods), yet this simply is not feasible - not even within experimental (i.e., non-observational) designs.

Response:

Our updated manuscript more clearly discusses why researchers might use less than ideal designs and why they are often not feasible (L425-435). We tried to make this point in the previous manuscript, but clearly our message was not well communicated – we hope our rephrasing and reframing of our manuscript has improved this.

Comment:

In line with this, I find the "if the design isn't extremely robust, it's hurting science"-narrative too harsh (see e.g. the opening lines of the abstract and the discussion section). In my view, there's nothing wrong with performing studies with less robust designs, just as it is perfectly valued to perform a small-sample study, *as long as* you don't overclaim your findings. If you only have an "After"-design, or if you have a sample that is too small and/or too homogeneous or a convenience sample, there are still conclusions that one could draw from the study; however these, obviously, are much more limited than those that can be drawn from large scale RCTs.

Response:

Our updated manuscript has the opposite narrative. Indeed, the previous version of the manuscript did not aim to have this narrative that 'less-than-ideal designs are hurting science'. We have completely rephrased and reframed our discussion and language throughout to better communicate why researchers might use less than ideal designs (L156-160, L425-435), why studies using these designs are still of crucial importance to decision-makers (L437-468) and should try to be included in evidence synthesis using the new methods we present (L470-488). We wholeheartedly agree with the viewpoint of the reviewer on this topic.

Comment:

It might be a tedious task, but it would certainly add value to the manuscript if the authors study to what extend the authors of the 6,532 papers studied indeed overclaim their results. If the authors don't overclaim on a massive scale, the harm for science of such 'inferior designs' is more limited and the paper should then be toned down accordingly. (Note: also in this situation the results are certainly publishable in my view.)

Response:

We agree this would be very interesting, but would be an incredibly laborious exercise and is not within the scope of the updated manuscript. We no longer overclaim or talk about the harm of using inferior designs as we have completely reframed and rephrased the manuscript and the language we use. We hope our original message all along is better communicated now.

Comment:

In the discussion/implication section, I would make a subsection with implications for meta research. I agree with the authors that for meta analyses, the type of research design certainly matters and that one cannot simply average the results of a RCT with those of some observational study. However, as outlined in the previous, I'm reluctant to go as far as the authors with the implications for science in general.

Response:

We provide more of a discussion about these issues for meta-research, meta-analyses and evidence synthesis in general (L437-488).

Comment:

Finally, I want to thank the authors for the extensive amount of 'supplementary materials'. They did a good job of separating the main message (which is in the manuscript) from details that are only of interest to a small proportion of readers (such as meta-analysts; these materials are the supplementary materials). Due to time limitations on my side, I did not thoroughly review the supplementary materials.

Response:

We thank the reviewer for this comment and hope that the supplementary materials we include now still achieve this, including full code and data to replicate our analyses.

Reviewer #3:

Comment:

This paper presents an analysis of the prevalence of the use of various kinds of experimental designs in social science and environmental research. While the analyses are fine, the interpretation of the results has limitations.

I take issue with the claim that they have demonstrated that the designs have “severely compromised the results” (line 393). They quantified the deviations. But they did not indicate how often the overall conclusion of the study was changed. We often are not interested in the precise parameter estimate, but rather the general direction of the effect.

Response:

We agree this was overstating our results and this is no longer present in the updated manuscript. We have rephrased and reframed the paper to address concerns by all reviewer that we were advocating against using these less-than-ideal designs. On the contrary, we were arguing the opposite and it was not helpful to use language like this as it clearly distorted our true messages from being effectively communicated.

Comment:

While mention is made of meta-analysis, the authors do not grapple with the question of whether these less-than-perfect designs, while problematic as single studies, still provide robust information when combined into a meta-analysis. If the main problem with these designs is simply an increase in the variance of the results, then the mean effect may still be robust. That is not true if the results are biased, but nowhere in this paper do the authors directly address that issue, except by anecdotal mentions of particular changes.

Response:

This is an interesting point and we have provided some discussion of this (L475-479). The results of our hierarchical model suggest that less reliable designs still provide useful information – the average bias (value of σ) for most designs, except BA, was far smaller than the average ‘true’ effect size (σ_{β} ; L445-448). We agree that it’s likely that for a large number of studies that these study design biases may cancel out, although this cannot be assumed, especially when meta-analyses use small numbers of studies. Ultimately, it is outside the scope of this analysis to determine this and we still believe that this does not preclude the need to account for the greater uncertainty associated with different study designs.

Comment:

I am also leery of the suggestion of weighting evidence in a meta-analysis by the study design. What sort of weighting factor would you use? Do you weigh a BACI study as twice the value of a Before-After study? Three times? It seems impossible to know how to weigh them without knowing the “true” effect.

Response:

Using our new analyses we present a new methodology to do exactly this. We present a hierarchical

model that provides a principled approach to combining studies that vary greatly in their design based on our quantification of the reliability of each design (L634-663).

Comment:

Nowhere do the authors discuss some of the reasons that a less than ideal design is used. For both conservation and social questions, experimental manipulations may be unethical or logistically unfeasible (e.g., diversity patterns over landscapes). Does that mean that we can never study those questions? No. It just means that in any such study the limitations of the data need to be acknowledged.

Response:

Our updated manuscript more clearly discusses why researchers might use less than ideal designs and why they are often not feasible (L156-160, L425-435). We tried to make this point in the previous manuscript, but our message was not communicated well – we hope our rephrasing and reframing of our manuscript has improved this. We completely agree with the viewpoint of the reviewer.

Comment:

One of the main points of the paper seems to be that a bigger sample size is better. This is hardly news. Importantly, the effects are all in terms of a percentage decrease of the original sample size. But a decrease of 50% when you start from 200 is not the same as when you start from 20, as statistical power is not linear with sample size. On the other hand, the authors make a point about the percentage of randomized control trials that had a different sign (line 320) based on a sample size of 5.

Response:

Our updated manuscript does not compare the results of study designs based on their sample sizes as this overcomplicated the previous analyses and was not the most important result. Our new analyses are substantially improved (L550-632).

Comment:

Title: As the number of such designs in other arenas is not quantified, the word “more” should be deleted from the title.

Response:

The title has been changed and better reflects our key results: ‘ ‘.

Reviewers' comments, second round:

Reviewer #1 (Remarks to the Author):

The authors have gone to a lot of effort to address reviewer comments, leading to a much-improved manuscript. The analysis is much better and results more intuitive, and the use of a mixed model to quantify bias (connecting equations 1 and 3 is a nice idea).

The main concern at this stage is the communication of, and indeed the method of implementation of, the variance components models (Table 2). At times the method was not communicated well, e.g. line 356 has a discussion of the value of λ , where λ has not been introduced at all.

There is a major caveat with interpretation of the results as "bias", because (as stated later) the true value of the effect size is unknown, so it is not known what truth actually is. This makes it impossible to say whether 0.003 is a better value than 0.03, and the discussion of these results needs to be tempered considerably to reflect this. In the special case of randomised trials, however, if randomisation was undertaken correctly, we can assume that BACI CA (and in fact CI) has a bias of zero. So one approach would be to fix the BACI effect at zero and estimate "relative bias", how far estimates tended to be above or below the BACI estimate. For lack of a better "gold standard", BACI-CA could be used in a similar way for non-randomised trials, but with more cautious interpretation.

Also the authors argue for leaving weighting out of their meta-analysis, which may have been done in error through misinterpretation of what the weights do – their equation 3 is of the same form as equation 1, and the weights apply to the last term only (the noise variable), and the bias terms are estimated by the random effects. So it is not the case that weighting is only appropriate for bias free models (contrary to lines 473-474) and it would seem weighting is appropriate. Reading through the maths of their method however it looks like weights were in fact included on the noise variables (via the structure of Λ) so it perhaps the text and discussion is wrong rather than the model that was fitted?

There may have also been some misinterpretation of results, with a section of the discussion recommending DiD over CA, when some DiD results were quite concerning (in particular, the lack of correspondence with CI for randomised designs).

Other comments:

Line 243 accurately

The close correspondence of BACI-CA and CI in Figure 3, even without randomisation, was surprising and interesting! The close correspondence with randomisation is reassuring. The lack of correspondence between BACI-DiD and CI for randomised designs is concerning.

Lines 386-392: this claim is problematic for two reasons. Firstly, it is unclear where the evidence for it is coming from (e.g. σ was larger in Table 2 for DiD, this difference did not look significant anyway, it is also unclear that 0 means unbiased). Secondly, it does not logically follow that greater observed bias means greater assumption violations. Incidentally, DiD also assumes normality, and "no unmeasured confounders" is implicit in the parallel trends assumption, so the stated differences in assumptions do not seem correct.

When comparing DiD and ANCOVA, the strong alignment between R-CI and R-BACI on Figure 3 for ANCOVA, and the lack of any such alignment for DiD, suggests problems with DiD. Randomisation should remove the need for before measurements, as explained in the manuscript, by setting design bias to zero, hence we expect R-CI and R-BACI to align. The DiD results there were surprising but they do caution strongly against DiD.

Lines 473-474 – if you weight observations inverse to variance, but include an observation-level random effect, this deals with bias (via the random effect) as well as noise (via weighting). Much of the discussion in the remainder of this paragraph would seem redundant.

Lines 596-606 – this could have been written more clearly, it is unclear what sort of GLMM was fitted to (what response type ended up being discussed much later, what the fixed effects were, what was mixed). It seems a lot of the time a GLM was fitted, with no random effect, but with a term for treatment (if BACI or CI), and time (for BACI or BA), and the family varying depending on the response type as explained below. So why not say that up front. Then add that if a hierarchical sampling design was used, with multiple samples in each site, a site random effect was added (if ≥ 10 sites with ≥ 2 samples per site).

Reviewer #2 (Remarks to the Author):

I've rarely seen a manuscript for which the revision was so much revised as this one. In my original review, I was positive on what the authors did, but criticized the way in which it was framed: it was too unclear, and too unfair towards situations in which RCTs are not possible.

This criticism now no longer holds, the revision is a **much** clearer description of the research study. I have no further points of criticism.

Casper Albers

Reviewer #3 (Remarks to the Author):

This paper presents an analysis of research designs in conservation and social sciences. This is the second time that I have reviewed this paper and the current version is much improved. The analysis is much better and the conclusions are more nuanced. Still, there are items that need to be addressed.

The major item is the statement that this paper provides a novel methodology. I am not a mathematical statistician and so cannot be definitive, but what is done appears to be a novel application of a standard analysis, not a truly new methodology. The analysis itself is valuable, but that is a different issue. I also am having a hard time seeing how it replaces standard meta-analysis. To some extent I can see how it might be an alternative method for parameter estimation. But meta-analyses go well beyond that.

The claims about their method being a replacement for standard meta-analyses need to be in its own manuscript where the two methodologies are placed in head-to-head competition with several datasets. There are certainly plenty of published meta-analyses in the ecology literature to choose from. In that regard, I disagree with the implication that meta-analyses are rare in environmental biology (lines 267-269). They may be rare in the conservation literature, but they have been embraced in ecology more broadly.

That discussion of meta-analysis also seems to misunderstand the way they are used in ecology when they say that they "can be confounded by variability in context or study populations." In medicine it is true that the goal is to look at a uniform set of studies because one wants to estimate a precise process that involves a unique factor. But in ecology, the goal is often to determine the existence of a general process or factor that manifests across many different contexts. That meta-analyses in ecology combine studies with different contexts and populations is a feature, not a bug. Nor is it the case that observation studies are systematically excluded from meta-analyses in ecology. It very much depends on the question being addressed.

In the discussion of potential biases of the analysis (lines 399-417), no mention is made about the fact that the estimators are not necessarily independent of each other. If multiple species are measured in the same study (line 584), they may be interacting with each other (e.g., an increase in the response of one species might lead to a decrease in the response of a different species).

Finally, I am having a problem with the title: "Quantifiably less biased study designs..." While it is true that the designs used are more biased, the authors also conclude that the magnitude of that bias was small (lines 445-448). The title should be rephrased to reflect that positive conclusion. Nor does this conclusion appear in the Abstract. It should be added.

Editorial fixes.

line 194: insert ":and _the_ use of a control group"

Table 1: Change the entries to percentages. It would also be less confusing to re-arrange the last five columns as: No overlap, Estimate of diff in magnitude, Significance of diff in magnitude, Estimate of diff in signs, and Significance of diff in signs. The first is a key result, which the others then break down as to cause. and it puts the estimate and the statistical significance of a given finding adjacent.

Table 2: It appears that a sigma got dropped from the caption.

lines 438-439: Currently phrased as a double negative. Rephrase as: "We believe that observation designs should be included in systematic reviews...."

Reviewer #1:**Comment:**

The authors have gone to a lot of effort to address reviewer comments, leading to a much-improved manuscript. The analysis is much better and results more intuitive, and the use of a mixed model to quantify bias (connecting equations 1 and 3 is a nice idea).

Response:

We are glad to hear that reviewer thinks the manuscript is much-improved and we appreciate their further helpful comments below.

Comment:

The main concern at this stage is the communication of, and indeed the method of implementation of, the variance components models (Table 2). At times the method was not communicated well, e.g. line 356 has a discussion of the value of lambda, where lambda has not been introduced at all.

Response:

We apologise for this poor communication. Lambda was explained in the Methods but not here, which we have now fixed (L360-370; Table 2).

Comment:

There is a major caveat with interpretation of the results as “bias”, because (as stated later) the true value of the effect size is unknown, so it is not known what truth actually is. This makes it impossible to say whether 0.003 is a better value than 0.03, and the discussion of these results needs to be tempered considerably to reflect this. In the special case of randomised trials, however, if randomisation was undertaken correctly, we can assume that BACI CA (and in fact CI) has a bias of zero. So one approach would be to fix the BACI effect at zero and estimate “relative bias”, how far estimates tended to be above or below the BACI estimate. For lack of a better “gold standard”, BACI-CA could be used in a similar way for non-randomised trials, but with more cautious interpretation.

Response:

We apologise that we have not communicated the variance component model effectively.

Our model uses the term σ to model the bias of study designs as a random effect across datasets; thus σ is the standard deviation of the bias. As the reviewer says, we cannot know what the actual bias is in each study, but if the bias is randomly distributed across studies and averages to be zero, we can gauge the magnitude of the bias through the variance component model. So a standard error (sigma) of 0.03 is better than 0.003, in the sense that the former design usually (but not always) has a smaller bias than the latter.

We have tried to add a reminder in the text when we discuss our results that we model bias as a random effect across the datasets – so sigma is the standard deviation of the bias (L341-344).

Comment:

Also the authors argue for leaving weighting out of their meta-analysis, which may have been done in error through misinterpretation of what the weights do – their equation 3 is of the same form as equation 1, and the weights apply to the last term only (the noise variable), and the bias terms are

estimated by the random effects. So it is not the case that weighting is only appropriate for bias free models (contrary to lines 473-474) and it would seem weighting is appropriate. Reading through the maths of their method however it looks like weights were in fact included on the noise variables (via the structure of Lambda) so it perhaps the text and discussion is wrong rather than the model that was fitted?

Response:

We again apologise for poor communication of our model and our argument. We do not advocate leaving weighting out of meta-analyses, rather that our model is a form of bias-adjusted meta-analysis that accounts for both bias and variance through weighting studies. We were suggesting in lines formerly 473-474 (now L470-482) that weighting simply by inverse variance or sample size is insufficient when studies suffer from different levels of bias. The difference between our model and previous bias-adjusted meta-analysis models is that we use empirically derived values of bias for each design, rather than eliciting expert opinions to rate the bias of each study design (L470-482).

In the article we deliberately tried not to be too technical or describe in detail how our model can be implemented for meta-analysis (e.g., using detailed formulae) to appeal to as broad a readership as possible. We believe a full description and implementation of bias-adjusted meta-analysis using our model is out of the scope of this paper, but still we would like to offer some comparison with the standard methods. We are planning to use Equation 1 for a future meta-analysis to demonstrate how this technique would work in practice, but we will provide a brief summary here. First, we gather all studies for the same true effect, obtain their effect size estimates, standard errors, and the type of study design. Then we plug them into the hierarchical model we used, where the effect size estimates share the same intercept (the true causal effect), a random effect term due to design bias (whose variance is estimated by the method we used), and a random effect term for statistical noise (whose variance is estimated by the reported standard error). We can then fit this model and estimate the shared intercept/causal effect. Heuristically, we can think of this as weighting the studies by both the bias of their design (σ) and their sampling variance (ϵ) (Equation 1 and Equation 2). We introduced Equation 3 to show how we accounted for correlated estimates because of the nested structure of our analyses (responses within different datasets), using the correlation matrix Ω and λ to account for systematic under/overestimation.

To make sure the reader better understands our arguments and methodology, we have rewritten the paragraph that likely confused the reviewer and hope our approach and arguments are now more clearly communicated (L470-482). We have also added a paragraph in the discussion to give a brief insight into how we envisage our novel approach being implemented in practice (L484-496).

Comment:

There may have also been some misinterpretation of results, with a section of the discussion recommending DiD over CA, when some DiD results were quite concerning (in particular, the lack of correspondence with CI for randomised designs).

Response:

Our results suggested that estimates from observational BACI DiD had smaller bias than BACI CA, but this was not statistically significant and in practice researchers should choose the estimator that's most appropriate and valid for their problem. We have tempered this discussion to reflect this (L390-396). There is also a known bracketing relationship between BACI DiD and CA (or lagged regression) which effectively means that both will typically yield the same sign, but may yield a point

estimate that is closer to the true magnitude depending on how well the underlying assumptions of each are met.

For randomised BACI DiD, it is clearer that this approach is unlikely to be useful because of its form since randomisation by definition should make the DiD estimator effectively redundant. Indeed, our results suggest that the extra variance introduced by the form of the DiD estimator is counterproductive to accurate quantification of the true effect.

Comment:

Line 243 accurately

Response:

We have corrected this typo (L241).

Comment:

The close correspondence of BACI-CA and CI in Figure 3, even without randomisation, was surprising and interesting! The close correspondence with randomisation is reassuring. The lack of correspondence between BACI-DiD and CI for randomised designs is concerning.

Response:

We noticed a labelling error in Figure 3 and the corrected Figure is now included in the manuscript (axes were inverted). We emphasise that Figure 3 was displaying t-statistics not point estimates. Therefore the fact R-BACI DiD corresponded poorly to R-CI is unsurprising, mainly due to the large number of results that were insignificant for R-BACI DiD, which can be attributed to the form of the DiD estimator which introduces additional, unnecessary noise into the estimate. Point estimates display a closer correspondence and we have now included a supplementary figure illustrating this (Supplementary Figure 1). The reason for the close correspondence of the R-BACI CA estimate is that the form of the CA estimator (using a lagged variable) makes little difference in a randomised setting and so is similar to CI.

Comment:

Lines 386-392: this claim is problematic for two reasons. Firstly, it is unclear where the evidence for it is coming from (e.g. sigma was larger in Table 2 for DiD, this difference did not look significant anyway, it is also unclear that 0 means unbiased). Secondly, it does not logically follow that greater observed bias means greater assumption violations. Incidentally, DiD also assumes normality, and “no unmeasured confounders” is implicit in the parallel trends assumption, so the stated differences in assumptions do not seem correct.

Response:

Our model uses the term sigma to model the standard deviation of bias across datasets. Therefore, sigma of zero = unbiased. We agree though that the difference was not significant and this was overselling our results; we now have a more cautionary finding that the underlying assumptions of DiD and CA should be carefully considered before applying either one (L390-396). In fact, DiD does not assume no unmeasured confounders; indeed, that is one of the advantages of the DiD estimator over CA, and hence there are differences in their underlying assumptions (see Ding, P. & Li, F. A

Bracketing Relationship between Difference-in-Differences and Lagged-Dependent-Variable Adjustment. *Political Analysis* 27, 605–615 (2019)).

Comment:

When comparing DiD and ANCOVA, the strong alignment between R-CI and R-BACI on Figure 3 for ANCOVA, and the lack of any such alignment for DiD, suggests problems with DiD. Randomisation should remove the need for before measurements, as explained in the manuscript, by setting design bias to zero, hence we expect R-CI and R-BACI to align. The DiD results there were surprising but they do caution strongly against DiD.

Response:

We emphasise that Figure 3 was displaying t-statistics not point estimates. The reason for the close correspondence of the R-BACI CA estimate is that the form of the CA estimator, including a pre-intervention measurement in the regression as a lagged variable, provided very little gain in model fit for the randomised datasets we looked at, and so has almost identical results to CI.

The fact R-BACI DiD corresponded poorly to R-CI is also unsurprising, mainly because of the large number of results that were insignificant for R-BACI DiD, which can be attributed to the form of the DiD estimator which introduces additional, unnecessary noise into the estimate. Hence our conclusion in L382-386. Point estimates display a closer correspondence and we have now included a supplementary figure illustrating this (Supplementary Figure 1).

Comment:

Lines 473-474 – if you weight observations inverse to variance, but include an observation-level random effect, this deals with bias (via the random effect) as well as noise (via weighting). Much of the discussion in the remainder of this paragraph would seem redundant.

Response:

We agree as this is exactly the approach we are implementing – we use sigma as a random effect to model the standard deviation in the bias across datasets, and epsilon as term to capture the noise. We have rewritten this paragraph to make it clearer what our approach does and how it is similar to bias-adjusted meta-analyses, except that crucially what we have done is to obtain empirical estimates of design bias rather than eliciting estimates of design bias from expert judgements (L470-482).

Comment:

Lines 596-606 – this could have been written more clearly, it is unclear what sort of GLMM was fitted to (what response type ended up being discussed much later, what the fixed effects were, what was mixed). It seems a lot of the time a GLM was fitted, with no random effect, but with a term for treatment (if BACI or CI), and time (for BACI or BA), and the family varying depending on the response type as explained below. So why not say that up front. Then add that if a hierarchical sampling design was used, with multiple samples in each site, a site random effect was added (if ≥ 10 sites with ≥ 2 samples per site).

Response:

We have now rearranged these paragraphs to more clearly communicate our methods in line with the reviewer's comments (L603-641).

Reviewer #2:

Comment:

I've rarely seen a manuscript for which the revision was so much revised as this one. In my original review, I was positive on what the authors did, but criticized the way in which it was framed: it was too unclear, and too unfair towards situations in which RCTs are not possible.

This criticism now no longer holds, the revision is a *much* clearer description of the research study. I have no further points of criticism.

Response:

We thank the reviewer and are glad that we have satisfied all their concerns.

Reviewer #3:

Comment:

This paper presents an analysis of research designs in conservation and social sciences. This is the second time that I have reviewed this paper and the current version is much improved. The analysis is much better and the conclusions are more nuanced. Still, there are items that need to be addressed.

Response:

We are pleased that the reviewer appreciates the work we have put in to improve the analysis.

Comment:

The major item is the statement that this paper provides a novel methodology. I am not a mathematical statistician and so cannot be definitive, but what is done appears to be a novel application of a standard analysis, not a truly new methodology. The analysis itself is valuable, but that is a different issue. I also am having a hard time seeing how it replaces standard meta-analysis. To some extent I can see how it might be an alternative method for parameter estimation. But meta-analyses go well beyond that.

Response:

We believe this depends on how you define methodology, but think a compromise would be to call this a novel approach. What is new about our work is that we have: 1.) used within-study comparisons on a large scale for both randomised and observational designs; and 2.) created a novel model to quantify differences in bias. We have tried to rewrite parts of the discussion to clearly show what we have done (L470-482), and reframed our methodology as a 'novel approach' (L374).

We do not believe we said our method should replace meta-analysis. Instead, what this offers is an alternative method of analysing studies that vary greatly in their design – something that traditional meta-analyses do not explicitly account for. Our model can be viewed as being similar to bias-adjusted meta-analysis, but what we have done is to approach this in a novel way by quantifying levels of bias empirically rather than quantifying bias via expert elicitation. To aid readability and prevent confusion in the future, we have reframed and reworded some of the paragraphs that may have led the reviewer to conclude we are recommending replacing meta-analyses with our model (L470-482).

Comment:

The claims about their method being a replacement for standard meta-analyses need to be in its own manuscript where the two methodologies are placed in head-to-head competition with several datasets. There are certainly plenty of published meta-analyses in the ecology literature to choose from. In that regard, I disagree with the implication that meta-analyses are rare in environmental biology (lines 267-269). They may be rare in the conservation literature, but they have been embraced in ecology more broadly.

Response:

We apologise for the confusion; again we did not mean that our model should replace meta-analysis. We offer an alternative method that is designed to be used when studies vary greatly in their design and is similar to bias-adjusted meta-analysis, but what we have done is to approach this in a novel way by quantifying levels of bias empirically rather than quantifying bias via expert elicitation.

We also believe there has also been some confusion around what we wrote in lines 267-269 (now 265-268); here we were not saying meta-analyses are rare, but rather the between-study comparisons of study designs are rare. That is, it is rare in our experience to see studies that aim to determine whether differences in the treatment effect are due to differences between study designs. This is only done sometimes in meta-analyses as a side analysis. Indeed, there is a rich literature on meta-analysis in ecology and is increasingly being used in conservation. To prevent confusion in the future, we have reframed and reworded this paragraph that may have led the reviewer to think we were implying meta-analyses are rare in environmental biology (L265-268).

Comment:

That discussion of meta-analysis also seems to misunderstand the way they are used in ecology when they say that they “can be confounded by variability in context or study populations.” In medicine it is true that the goal is to look at a uniform set of studies because one wants to estimate a precise process that involves a unique factor. But in ecology, the goal is often to determine the existence of a general process or factor that manifests across many different contexts. That meta-analyses in ecology combine studies with different contexts and populations is a feature, not a bug. Nor is it the case that observation studies are systematically excluded from meta-analyses in ecology. It very much depends on the question being addressed.

Response:

We thank the reviewer for raising the interesting point that meta-analysis in ecology often has different features and aims to those in medicine; indeed, it does depend on the question that is

being asked as to whether non-uniformity in studies is a 'bug' or a useful feature. We gave some thoughts to this and felt that a discussion of this is outside the scope of this paper.

Here we apologise again for not making the paragraph the reviewer refers to clearer as we were focusing on studies that aim to conduct between-study comparisons of study designs (i.e., those trying to determine whether differences in the treatment effect are due to differences between study designs). When that is the aim of the study then such analyses "can be confounded by variability in context or study populations." Therefore, within-study comparisons to understand the influence of study design on study findings are more reliable and less likely to be confounded. So our case was that there tend to be fewer studies in the environmental sciences that seek to determine whether differences in the treatment effect are due to differences between study designs, compared to medicine for example. We have tried to better explain this in the paragraph and remove the confusing reference to meta-analyses (L265-268).

We agree that observational studies often are not systematically excluded from meta-analyses in ecology, but are in other disciplines, and we want to make a broader point for wider approaches to evidence synthesis. Of course, this also depends on the philosophical standpoint of those undertaking the analysis. Whether disciplines rightly or wrongly exclude observational studies, our novel approach demonstrates the importance of accounting for possible biases introduced by these studies.

Comment:

In the discussion of potential biases of the analysis (lines 399-417), no mention is made about the fact that the estimators are not necessarily independent of each other. If multiple species are measured in the same study (line 584), they may be interacting with each other (e.g., an increase in the response of one species might lead to a decrease in the response of a different species).

Response:

Thank you for pointing out this very good point. We did try to model the correlations between different estimators for the same response in our hierarchical model (using omega the correlation matrix in Equation 3), but we did not account for dependence across the responses/species. We have now made sure to explain this limitation (L406-410).

Comment:

Finally, I am having a problem with the title: "Quantifiably less biased study designs..." While it is true that the designs used are more biased, the authors also conclude that the magnitude of that bias was small (lines 445-448). The title should be rephrased to reflect that positive conclusion. Nor does this conclusion appear in the Abstract. It should be added.

Response:

We agree a better title is needed to fairly represent our findings while being interesting and informative. We have decided to rename the study as 'Design matters: quantifying the prevalence and bias of study designs in the environmental and social sciences'. We believe that the result that biases were generally small for observational designs needs to be tempered to account for the uncertainty in our non-randomised dataset model, as we could be underestimating the magnitude of

these biases (L400-403). Therefore, we suggest that carefully designed observational designs (particularly controlled rather than uncontrolled observational designs), can still provide useful information and therefore have a place in evidence synthesis (L443-445). Unfortunately we do not believe it is possible to make this nuanced point in the abstract given the 150 word limit.

Comment:

line 194: insert “:and _the_ use of a control group”

Response:

We have added the word ‘the’ to correct this (L194).

Comment:

Table 1: Change the entries to percentages. It would also be less confusing to re-arrange the last five columns as: No overlap, Estimate of diff in magnitude, Significance of diff in magnitude, Estimate of diff in signs, and Significance of diff in signs. The first is a key result, which the others then break down as to cause. and it puts the estimate and the statistical significance of a given finding adjacent.

Response:

We have changed the entries to percentages and rearranged the last five columns as the reviewer has said (Table 1).

Comment:

Table 2: It appears that a sigma got dropped from the caption.

Response:

There were some mathematically formatted terms here that were accidentally deleted by MS Word and we have now added these back in (Table 2).

Comment:

lines 438-439: Currently phrased as a double negative. Rephrase as: “We believe that observation designs should be included in systematic reviews....”

Response:

We have rephrased this to remove the double negative and to make a more nuanced point here (L443-445).

Reviewers' comments, third round:

Reviewer #1 (Remarks to the Author):

The paper is much improved, a few minor comments:

line 470 "Our novel hierarchical model" - but in a previous response it was agreed that while this may be an original application of mixed models, it is not in itself a "new model"

line 341 The hierarchical Bayesian model comes out of nowhere, maybe rephrase as "We modelled study design bias using a random effect across datasets in a hierarchical Bayesian model; σ is the standard deviation of this bias term, and assuming..."

line 408 "dependent" (not "independent")

Reviewer #3 (Remarks to the Author):

This is now the third time that I have reviewed this manuscript. The authors have done an excellent job in responding to the previous criticisms. I have no further concerns. This paper represents a substantial advance on current methods for combining data from multiple studies.

Reviewer #1:

Comment:

line 470 "Our novel hierarchical model" - but in a previous response it was agreed that while this may be an original application of mixed models, it is not in itself a "new model"

Response:

We have removed the reference to the hierarchical model being novel.

Comment:

line 341 The hierarchical Bayesian model comes out of nowhere, maybe rephrase as "We modelled study design bias using a random effect across datasets in a hierarchical Bayesian model; σ is the standard deviation of this bias term, and assuming..."

Response:

We have now rephrased this line as the reviewer suggested.

Comment:

line 408 "dependent" (not "independent")

Response:

We have changed 'independent' to 'dependent'